# Copycats: the many lives of a publicly available medical imaging dataset

**Amelia Jiménez-Sánchez**[1]     **Natalia-Rozalia Avlona**[2]     **Dovile Juodelyte**[1]     **Théo Sourget**[1]
**Caroline Vang-Larsen**[1]     **Anna Rogers**[1]     **Hubert Dariusz Zając**[2]     **Veronika Cheplygina**[1]
[1]IT University of Copenhagen     [2]University of Copenhagen
{amji,vech}@itu.dk

## Abstract

Medical Imaging (MI) datasets are fundamental to artificial intelligence in healthcare. The accuracy, robustness, and fairness of diagnostic algorithms depend on the data (and its quality) used to train and evaluate the models. MI datasets used to be proprietary, but have become increasingly available to the public, including on community-contributed platforms (CCPs) like Kaggle or HuggingFace. While open data is important to enhance the redistribution of data's public value, we find that the current CCP governance model fails to uphold the quality needed and recommended practices for sharing, documenting, and evaluating datasets. In this paper, we conduct an analysis of publicly available machine learning datasets on CCPs, discussing datasets' context, and identifying limitations and gaps in the current CCP landscape. We highlight differences between MI and computer vision datasets, particularly in the potentially harmful downstream effects from poor adoption of recommended dataset management practices. We compare the analyzed datasets across several dimensions, including data sharing, data documentation, and maintenance. We find vague licenses, lack of persistent identifiers and storage, duplicates, and missing metadata, with differences between the platforms. Our research contributes to efforts in responsible data curation and AI algorithms for healthcare.

## 1   Introduction

Datasets are fundamental to the fields of machine learning (ML) and computer vision (CV), from interpreting performance metrics and conclusions of research papers to assessing adverse impacts of algorithms on individuals, groups, and society. Within these fields, medical imaging (MI) datasets are especially important to the safe realization of Artificial Intelligence (AI) in healthcare. Although MI datasets share certain similarities to general CV datasets, they also possess distinctive properties, and treating them as equivalent can lead to various harmful effects. In particular, we highlight three properties of MI datasets: (i) de-identification is required for patient-derived data; (ii) since multiple images can belong to one patient, data splits should clearly differentiate images from each patient; and (iii) metadata containing crucial information such as demographics or hospital scanner is necessary, as models without this information could lead to inaccurate and biased results.

In the past, MI datasets were frequently proprietary, confined to particular institutions, and stored in private repositories. In this particular setting, there is a pressing need for alternative models of data sharing, documentation, and governance. Within this context, the emergence of Community-Contributed Platforms (CCPs) presented a potential for the public sharing of medical datasets. Nowadays, more MI datasets have become publicly available and are hosted on open platforms such as grand-challenges[1], or CCP - including companies like Kaggle or HuggingFace.

---

[1]https://grand-challenge.org

38th Conference on Neural Information Processing Systems (NeurIPS 2024) Track on Datasets and Benchmarks.

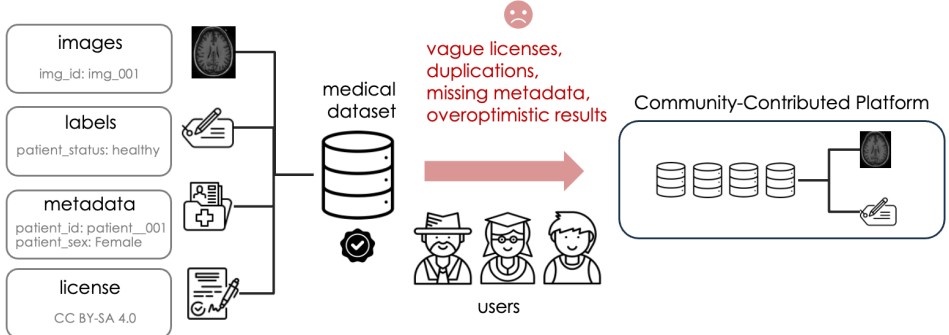

Figure 1: A Medical Imaging (MI) dataset containing images, labels, metadata (patient id, patient sex, etc.), and license (left). After user interaction, on Community-Contributed Platforms we find duplicated data, missing licenses and metadata, which can lead to overoptimistic results (right).

Although the increasing availability of MI datasets is generally an advancement for sharing and adding public value, it also presents several challenges. First, according to the FAIR (Findable, Accessible, Interoperable, Reusable) guiding principles for scientific data management and stewardship [122], (meta)data should be released with a clear and accessible data usage license and should be permanently accessible. Second, tracking dataset versions is becoming increasingly difficult, especially when publications use derived versions [89] or the citation practices are not followed [109]. This hampers the analysis of usage patterns to identify possible ethical concerns that might arise after releasing a dataset [33], potentially leading to its retraction [89, 60]. To mitigate the harms associated with datasets, ongoing maintenance, and stewardship are necessary [89]. Lastly, rich documentation is essential to avoiding over-optimistic and biased results [15, 68, 32, 125, 86], attributed to a lack of meta-data in MI datasets, such as information linking images to specific patients and their demographics. Documentation needs to reflect all the stages in the dataset development cycle, such as acquisition, storage, and maintenance [51, 40]. Although CCPs offer ways to enhance the redistribution of data's public value and alleviate some of these problems providing structured summaries, we find that the current CCP governance model fails to uphold the quality needed and recommended practices for sharing, documenting, and evaluating MI datasets.

In this paper, we investigate MI datasets hosted on CCPs, particularly how they are documented, shared, and maintained. First, we provide relevant background information, highlighting the differences between open MI and CV datasets, especially in the potential for harmful downstream effects of poor documentation and distribution practices (Section 2.1). Second, we present key aspects of data governance in the context of ML and healthcare, specifically affecting MI datasets (Section 2.2). Third, we analyze *access, quality* and *documentation* of 30 popular datasets hosted on CCPs (10 medical, 10 computer vision, and 10 natural language processing). We find issues across platforms related to vague licenses, lack of persistent identifiers and storage, duplicates, and missing metadata (Section 3). We discuss the limitations of the current dataset management practices and data governance on CCPs, provide recommendations for MI datasets, and conclude with a discussion of limitations of our work and open questions (Section 4).

## 2 Background

### 2.1 Characteristics of medical imaging datasets

**Anatomy of a medical imaging dataset.** A MI dataset begins with a collection of images from various imaging modalities, such as X-rays, magnetic resonance imaging (MRI), computed tomography (CT) scans, and others. The scans are often initially captured for a clinical purpose, such as diagnosis or treatment planning, and are associated with a specific patient and their medical data. The scans might undergo various processing steps, such as denoising, registration (aligning different scans together), or segmentation (delineating anatomical structures or pathologies). Clinical experts might then associate the scans with additional information, *e.g.*, free text reports or diagnostic labels.

A collection of scans and associated annotations, *i.e.*, a MI dataset, might be later used for the purpose of training and evaluating ML models supporting the work of medical professionals [133, 115]. However, before a dataset is "ready" for ML, further steps are required [123], including cleaning (for example, removing scans that are too blurry), sampling (for example, only selecting scans with a particular disease), and removing identifying patient information. Additional annotations, not collected during clinical practice, may be required to train ML models, *e.g.*, organ delineations for patients not undergoing radiotherapy. These annotations might be provided by clinical experts, PhD students, or paid annotators at tech companies.

**Not just "small computer vision"!**    While MI datasets share some similarities with general CV datasets, they also have unique properties. The diversity of image modalities and data preprocessing needed for each specific application is vast. For instance 3D images from modalities like MRI can vary significantly depending on the sequence used. For example, brain MRI sequences (T1-weighted, T2, FLAIR, etc.), are designed to emphasize different brain structures, offering specific physiological and anatomical details. Whole-slide images of histopathology are extremely large (gigapixel) images, making preprocessing both challenging and essential for accurate analysis. A crucial part of this process is stain normalization, which standardizes color variations caused by different staining processes, ensuring consistency across slides for more reliable analysis and comparison [29]. We refer interest readers in knowing more about preparing MI data of different modalities for ML for example to [123, 67].

Nevertheless, the complexity of medical image data above is often reduced to a collection of ML-library-ready images and labels. Yet treating MI datasets as equivalent to benchmark CV datasets is problematic and leads to harmful effects, also termed *data cascades* by [102]. *Data cascades* can lead to degraded model performance, reinforce biases, increase maintenance costs, and reduce trust in AI systems. These problems often stem from poor data quality, lack of domain expertise, and insufficient documentation, which become increasingly difficult to correct once models are deployed.

First, unlike traditional CV datasets, medical images often require de-identification processes to remove personally identifiable data, which are more complex than complete anonymization. Certain attributes like sex and age, need to be preserved for clinical tasks. These attributes are typically included in an "original release" of MI datasets, they might be removed later in a dataset's lifecycle. For example, when medical datasets are shared on CCPs, often only the input desired by ML practitioners remains: inputs (images) and outputs (disease labels), as shown in Figure 1.

Second, MI datasets often include multiple images associated with a single patient. This can occur if a patient has multiple skin lesions, follow-up chest X-rays, or 3D scans split into 2D images. If images from the same patient end up in both training and test data, reported results may be overly optimistic as classifiers memorize patients rather than disease characteristics. Therefore, data splitting at the patient level is crucial to avoid model overfitting. While this practice is common in the MI community, it may be overlooked if datasets are simply shared as a general CV dataset.

Third, MI datasets should contain metadata about patient demographics. Several studies have shown how demographic data may alleviate systematic biases and impact disease classification performance in chest X-rays [68, 106] and skin lesions [4]. These datasets are often the subject of research on bias and fairness because they include variables for age and sex or gender (typically not described which). However, many MI datasets lack these variables, possibly due to removal in a ML-ifying step rather than actual anonymization. Unlike CV datasets where bias can be identified by annotating individuals in the images based on their gender expression [131], such information is often unrecoverable from medical images. Additionally, images may be duplicated; see, *e.g.*, [20] for an analysis of the ISIC datasets, with overlaps between versions and duplication of cases between training and test sets.

Finally, MI datasets should include metadata about the origin of scans. Lack of such data may lead to "shortcuts" and other systematic biases. For example, if disease severity correlates with the hospital where the scans were made (general *vs.* cancer clinic), a model might learn the clinic's scanner signature as a shortcut for the disease [27]. In other words, the shortcut is a spurious correlation between an artifact in the image and the diagnostic label. Some examples of shortcuts include patient position in COVID-19 [32], chest drains in pneumothorax classification [86, 57], or pen marks in skin lesion classification [125, 15, 26]. High overall performance can hide biases in benchmark evaluations serving underrepresented groups. This cannot be detected without appropriate metadata.

**Evolution of MI datasets.** Historically, MI datasets were often proprietary, limited to specific institutions, and held in private repositories. Due to the privacy concerns and high cost associated with expert annotations, the sizes of MI datasets were quite small, often in the tens or hundreds of patients, which limited the use of machine learning techniques. Over the years some datasets have become publicly available and increased in size, for example, INBreast [82] and LIDC-IDRI [8] with thousands, and some even with tens or hundreds of thousands of patients, like chest X-ray datasets (NIH-CXR14 [118], MIMIC-CXR [58], CheXpert [52]), and skin lesions datasets (ISIC [22, 26]). To augment the dataset's size and alleviate the high cost of annotation, dataset creators used NLP techniques to automatically extract labels from medical reports, at the expense of annotation reliability [85]. Lately, advancements in large language models have redirected the attention of the MI community towards multi-modal models with both text and images. MI datasets are increasingly used to benchmark general ML and CV research.

Next to unreliable annotations, publicly available MI datasets have increasingly exhibited biases and spurious correlations or shortcuts. For example, several studies have shown differences in the performance of disease classification in chest X-rays [68, 106] and skin lesions [4] according to patient demographics. Spurious correlations could also bias results, like chest drains affecting pneumothorax classification [86, 57], or pen marks influencing skin lesion classification [125, 15].

Thus, MI datasets need to be updated, similar to ML datasets which may be audited or retracted [89, 60, 33]. These practices are currently not formalized. Even tracking follow-up work on specific datasets is challenging due to the lack of stable identifiers and proper citations [109]. Data citation is crucial for making data findable and accessible, offering persistent and unique identifiers along with metadata [46, 23]. Instead, datasets are often referenced by a mix of names or URLs in footnotes. When datasets are updated, it often occurs informally. For instance, the LIDC-IDRI website uses red font size to signal errors and updated labels, while the NIH-CXR14 website hosts derivative datasets and job announcements. There exist some systematic reviews of MI datasets [31, 119, 72], however, this is not a common practice. Furthermore, changes to datasets cannot be captured with traditional literature-based reviews.

## 2.2  Data management practices in the medical imaging context

**Data governance, documentation, and data hosting practices.** Data governance is a nebulous concept with evolving definitions that vary depending on context [6]. Some definitions relate to strategies for data management [6], others to formulation and implementation of data stewards' responsibilities [30]. The goals of data governance are ensuring the quality and proper use of data, meeting compliance requirements, and helping utilize data to create public value [54]. In the context of research data, a relevant initiative is the FAIR guiding principles for scientific data management and stewardship [122]. These principles ensure that data is *findable*, easily located by humans and computers with unique identifiers and rich metadata; *accessible*, retrievable using standard protocols; *interoperable*, *i.e.* it uses shared, formal languages and standards for data and metadata; *reusable*, clearly licensed, well-documented, and meeting community standards for future use. The CARE principles for Indigenous data governance [19] complement FAIR, ensuring that data practices respect Indigenous sovereignty and promote equitable outcomes.

At this point there are multiple studies proposing and discussing data governance models for the ML community and its various subfields [54, 65, 81, 55]. For example, a model proposed for radiology data in [81] is based on the principles of stewardship, ownership, policies, and standards[2]. A relevant line of research is the efforts by ML researchers raising awareness about the importance of dataset documentation and proposing guidelines, frameworks, or datasheets [48, 12, 94, 40, 34, 51]. Despite the existence of many excellent data governance and documentation proposals, the challenge lies in the implementation of their principles. One of the most common ways to share and manage ML datasets is for the developers to host data themselves upon release, often on platforms like GitHub

---

[2]*Stewardship* considers accountability for data management and involves establishing roles and responsibilities to ensure data quality, security, and protection. *Ownership* identifies the relationship between people and their data and is distinct from stewardship in that the latter maintains ownership by accountable institutions. *Policies*: considers organizational rules and regulations overseeing data management. These rules are, for example, to protect data from unauthorized access or theft, as well as to consider the impact of data, ethics, and legal statutes. *Standards*: specific criteria and rules informing proper handling, storage, and maintenance of data throughout its lifecycle.

or personal websites. Coupled with the lack of agreement on which data governance principles should be implemented, this means the lack of standardization and consistent implementation of any data governance frameworks. Another common solution is to host data on CCPs such as Kaggle or HuggingFace. CCPs resemble a centralized structure, but they heavily rely on user contributions for both sourcing datasets and governance. For example, while HuggingFace developed the infrastructure for providing data sheets and model cards, this is not enforced, and many models or datasets are uploaded with minimal or no documentation as we show in this paper. In theory, Wikimedia sets a precedent for some standardization in a highly collaborative and largely self-regulated project [37], but data documentation is a more challenging task for someone other than the original contributor, and it is unfortunately also a less rewarded and prestigious task in the ML community [102].

**Data governance for healthcare data.** Health data is considered a high-risk domain due to the sensitive and personal nature of the information it contains. Thus, releasing MI datasets poses regulatory, ethical, and legal challenges, related to privacy and data protection [97]. Furthermore, patient demographics could include information about vulnerable populations such as children [81], or underrepresented or minority groups, including Indigenous peoples [45]. To take into account underrepresented populations and ensure transparency, ML researchers entering medical applications should adhere to established healthcare standards. These include data standards like FHIR[3] [9], which allows standardized data access using REST architectures and JSON data formats. Additionally, they should follow standard reporting guidelines such as TRIPOD[4] [24] and CONSORT[5] [103], which are now being adapted for ML applications [10, 25]. These guidelines set reasonable standards for transparent reporting and help communicate the analysis method to the scientific community. Various guidelines exist for preparing datasets for ML [123], including the FUTURE-AI principles [71], which aim to foster the development of ethically trustworthy AI solutions for clinical practice.

A key challenge in MI, as in ML in general, is the sparsity of consistent implementation of data governance principles. For example, Datasheets [40] was adopted for CheXpert in [39], but such practices remain uncommon, as our study shows. Besides the common CCPs and self-hosting options discussed above, some MI datasets are also hosted on platforms like grand-challenges, Zenodo [36], Physionet [44], and Open Science Framework [38]. Of these platforms, only Physionet consistently collects detailed documentation for the datasets. These platforms are not integrated into the commonly used ML libraries and hence tend to be less well-known in the ML community.

## 3   Findings

**Study setup.** We aim to promote better practices in the context of MI datasets. For that, we investigate dataset sharing, documentation, and hosting practices for the 30 most cited CV, NLP, and MI datasets by selecting top-10 datasets for each field by querying Papers with Code with "Images", "Text", and "Medical" in the Modality field. We include CV and NLP in the comparison because MI is often inspired by these other ML areas, and is where data governance have recently received more attention. We were expecting to find MI datasets like BraTs [80], ACDC [13], etc. in the list, since we thought they are commonly used, but we decided to leverage Papers with Code to retrieve datasets in a systematic way. In Table 1, we show the original source where each dataset is hosted, the distribution terms (for use, sharing, or access), the license, and other platforms where the datasets can be found. In particular, we investigate dataset distribution on CCPs such as Kaggle and HuggingFace (HF), and regulated platforms (RP) such as Tensorflow (TF), Keras, and PyTorch. Furthermore, we analyze the Kaggle and HuggingFace datasets by automatically extracting the documentation elements associated with each dataset using their APIs. We obtain the parameters in data cards as specified in their dataset creation guides [1, 3].

**Lack of persistent identifiers, storage, and clear distribution terms.** In Table 1, we observe that CV and NLP datasets are mostly hosted on authors or university websites. This is not aligned with the FAIR guiding principles. In contrast, MI datasets are hosted on a variety of websites: university, grand-challenge, or PhysioNet. We find some examples of datasets that follow the FAIR principles [122], like HAM10000 with a persistent identifier, or MIMIC-CXR stored in PhysioNet [44], which offers

---

[3]FHIR: Fast Health Interoperability Resources
[4]TRIPOD: Transparent Reporting of a multivariable prediction model for Individual Prognosis Or Diagnosis
[5]CONSORT: Consolidated Standards of Reporting Trials

| | Dataset | Original hosting source | Distribution terms (use, access, sharing) | License | Please cite this paper | Kaggle | HF | TF | Keras | PyTorch |
|---|---|---|---|---|---|---|---|---|---|---|
| | | | | | | CCP | | RP | | |
| 1 | CIFAR-10 [63] | □ cs.toronto.edu/kriz/cifar.html | | □ Unspecified | Yes | ✓ | ✓ | ✓ | ✓ | ✓ |
| 2 | ImageNet [101] | □ image-net.org | Terms of access | □ Unspecified | Yes | ✓ | ✓ | ✓ | | ✓ |
| 3 | CIFAR-100 [63] | □ cs.toronto.edu/kriz/cifar.html | | □ Unspecified | Yes | ✓ | ✓ | ✓ | ✓ | ✓ |
| 4 | MNIST [70] | □ yann.lecun.com/exdb/mnist/ | | □ Unspecified | Yes | ✓ | ✓ | ✓ | ✓ | ✓ |
| 5 | SVHN [83] | □ ufldl.stanford.edu/housenumbers/ | | □ Unspecified | Yes | ✓ | ✓ | ✓ | | ✓ |
| 6 | CelebA [75] | □ mmlab.ie.cuhk.edu.hk/projects/CelebA.html | Agreement | □ Unspecified | Yes | ✓ | ✓ | ✓ | | ✓ |
| 7 | Fashion-MNIST [126] | △ github.com/zalandoresearch/fashion-mnist | | ● MIT | Yes | ✓ | ✓ | ✓ | ✓ | ✓ |
| 8 | CUB-200-2011 [116] | □ vision.caltech.edu/datasets/cub_200_2011/ | | □ Unspecified | Yes | ✓ | ✓ | ✓ | | |
| 9 | Places [132] | □ places.csail.mit.edu | | △ (C), ● CC-BY | Yes | ✓ | | ✓ | | ✓ |
| 10 | STL-10 [21] | □ cs.stanford.edu/ acoates/stl10 | | □ Unspecified | Yes | ✓ | ✓ | ✓ | ✓ | ✓ |
| 1 | GLUE [117] | △ gluebenchmark.com/ | | See original datasets | Yes | ✓ | ✓ | ✓ | | ✓ |
| 2 | SST [108] | □ nlp.stanford.edu/sentiment/ | | □ Unspecified | Yes | ✓ | ✓ | ✓ | | ✓ |
| 3 | SquAD [96] | □ rajpurkar.github.io/SQuAD-explorer/ | | ● CC-BY-SA 4.0 | No | ✓ | ✓ | ✓ | | ✓ |
| 4 | MultiNLI [124] | □ cims.nyu.edu/sbowman/multinli/ | | ● Various CC | Yes | ✓ | ✓ | ✓ | | ✓ |
| 5 | iMDB reviews [77] | □ ai.stanford.edu/amaas/data/sentiment/ | | □ Unspecified | Yes | ✓ | ✓ | ✓ | | ✓ |
| 6 | VQA [7] | □ visualqa.org/ | Terms of use | △ (C), ● CC-BY | Yes | ✓ | ✓ | ✓ | | ✓ |
| 7 | SNLI [16] | □ nlp.stanford.edu/projects/snli | | ● CC-BY-SA 4.0 | Yes | ✓ | ✓ | ✓ | | ✓ |
| 8 | Visual Genome [62] | ● homes.cs.washington.edu/[...]visualgenome | | ● CC-BY 4.0 | Yes | ✓ | ✓ | ✓ | | |
| 9 | QNLI | △ gluebenchmark.com/ - derived from SQUAD | | □ Unspecified | No | ✓ | ✓ | ✓ | | ✓ |
| 10 | Natural Questions [64] | △ ai.google.com/research/NaturalQuestions | | ● CC-SA 3.0 | No | ✓ | ✓ | ✓ | | |
| 1 | CheXpert [52] | □ stanfordmlgroup.github.io/competitions/chexpert/ | Research Use | □ Unspecified | Yes | ✓ | | ✓ | | |
| 2 | DRIVE [110] | ● drive.grand-challenge.org | | □ Unspecified | No | ✓ | | | | |
| 3 | fastMRI [59] | □ fastmri.med.nyu.edu | Sharing Agreement | □ Unspecified | Yes | | ✓ | | | |
| 4 | LIDC-IDRI [8] | ● wiki.cancerimagingarchive.net/[...]pageId=[...] | TCIA Data Usage | ● CC-BY-3.0 | Yes | ✓ | | | | |
| 5 | NIH-CXR14 [118] | △ nihcc.app.box.com/v/ChestXray-NIHCC | | □ Unspecified | Yes | ✓ | | | | |
| 6 | HAM10000 [113] | ● dataverse.harvard.edu/[...]persistentId=doi[...] | Use Agreem. | ● CC-BY-NC-4.0 | Yes | ✓ | ✓ | | | |
| 7 | MIMIC-CXR [58] | ● physionet.org/content/mimic-cxr/2.0.0/ | Phys. Use Ag. 1.5.0 | ● PhysioNet 1.5.0 | Yes | ✓ | ✓ | | | |
| 8 | Kvasir-SEG [56] | △ datasets.simula.no/kvasir-seg/ | Terms of use | □ Unspecified | Yes | ✓ | ✓ | | | |
| 9 | STARE [49] | □ cecas.clemson.edu/ ahoover/stare/ | | □ Unspecified | Yes | ✓ | | | | |
| 10 | LUNA [105] | ● luna16.grand-challenge.org | | ● CC-BY-4.0-DEED | Yes | ✓ | | | | |

Table 1: Original hosting source, distribution terms, license, and alternative hosting platforms for the top-10 datasets from Papers with Code for the modality "Images" (top), "Text" (middle), and "Medical" (bottom). Hosting: □ author or university website; △ open but not permanent access; ● open and permanent access. License: □ unspecified; △ copyright; ● MIT, CC or Physionet. CCP: Community-Contributed Platforms. RP: Regulated Platforms. HF: HuggingFace, TF: TensorFlow.

permanent access to datasets with a Digital Object Identifier (DOI). Without a persistent identifier and storage, access to the (meta)data is uncertain, which is problematic for reproducibility. Regarding dataset distribution on other platforms, we observe that CV and NLP datasets are available both on CCP and RP. This is not the case for MI datasets, which are not commonly accessible on RP, but some of them are on CCP. A possible reason could be that RPs are mindful of the licensing or distribution terms of the datasets, or that their infrastructure does not easily accommodate MI datasets.

Licenses or terms of use represent legal agreements between dataset creators and users, yet we observe in Table 1 (top) that the majority of the most used CV datasets were not released with a clear license or terms of use. This observation aligns with [76], who report that over 70% of widely used dataset hosting sites omit licenses, indicating a significant issue of misattribution. Regarding MI datasets, we observe in Table 1 (bottom) that less than half of the most used datasets were released with a license. Even if Papers with Code shows that DRIVE dataset is under a CC-BY-4.0 license, the dataset creators confirmed to us that they did not specify any license when they released it.

**Duplicate datasets and missing metadata on CCPs.**    We present a case study of "uncontrolled" spread of skin lesion datasets from the ISIC archive [2], focused on the automated diagnosis of melanoma from dermoscopic images. These datasets originated from challenges held between 2016 and 2020 at different conferences. Each challenge introduced a new compilation of the archive data, with potentially overlapping data with previous instances [20]. The ISIC datasets can be downloaded from their website[6], and depending on the dataset, there are various licenses like CC-0 and CC-BY-NC, and researchers are requested to cite the challenge paper and/or the original sources of the data (HAM10000 [113], BCN20000 [47], MSK [22]).

As of May 2024, there are 27 datasets explicitly related to ISIC on HuggingFace. Some of these datasets are preprocessed (*e.g.*, cropped images), others provide extra annotations (*e.g.*, segmentation masks). Kaggle has a whopping 640 datasets explicitly related to ISIC. While the size of the original ISIC datasets is 38 GB, Kaggle stores 2.35 TB of data (see Figure 2). Several highly downloaded versions ($\approx$13k downloads) lack original sources or license information. This proliferation of duplicate datasets not only wastes resources but also poses a significant impediment to the reproducibility of

---

[6]https://challenge.isic-archive.com/data/

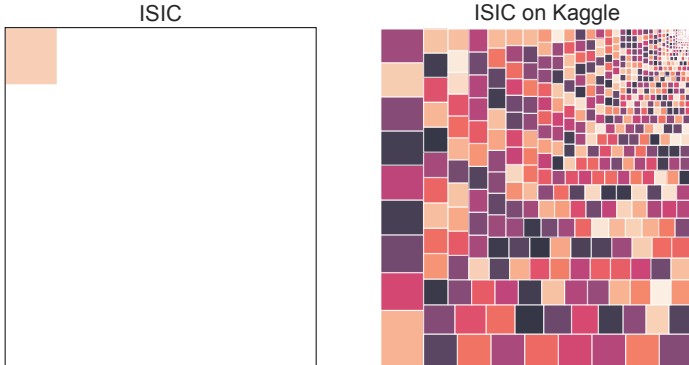

Figure 2: Representation of the storage size for ISIC (skin lesion) datasets. While the ISIC website hosts a total of 38 GB of data (left), on Kaggle there are a total of 640 datasets related to ISIC (some preprocessed, other with additional annotations), that sum up to 2.35 TB of data (right). Each block on the (right) represents a single instance of ISIC-derived dataset on Kaggle. Block size represents dataset size. Data was retrieved on May 15, 2024.

research outcomes. Besides ISIC, on Kaggle we find other examples of unnecessary duplication of data, see Table 3 of the Supplementary Material for details.

After this finding about ISIC, we examined several other datasets for duplication. On Kaggle, we find 350 datasets related to BraTS (Brain Tumor Segmentation) [80], and 24 datasets of INBreast [82], one of them with the rephrased description "I'm just uploading here this data as a backup". Additionally, there are 10 instances of PAD-UFES-20 [88] (also a skin lesion dataset, one instance actually contains data from ISIC). The ACDC (Automated Cardiac Diagnosis Challenge) dataset [13] consists of MRIs, while ACDC-LungHP (Automatic Cancer Detection and Classification in Lung Histopathology) dataset [73, 74] contains histopathological images. On Kaggle, we find an example of a dataset titled "ACDC lung" that contains cardiac images.

The lack of documentation for *all* ML, not just MI datasets, hampers tracking their usage, potentially violates sharing agreements or licenses, and hinders reproducibility. Additionally, due to the characteristics of MI datasets, models trained on datasets missing metadata could result into overoptimistic performance due to data splits mixing patient data, or bias [68] or shortcuts [86, 125]. We therefore reviewed the documentation on the original websites and related papers for the MI datasets, and found that patient data splits were clearly reported for 6 out of 10 datasets – "clearly reported" means that a field like "patient_id" was provided for each case. However, tracking whether data splits are defined at the subject level for duplicates on CCPs is challenging, as the relevant information is not always in the same location. One must examine the file contents (often requiring downloading the entire dataset) of each duplicated dataset to determine if a field like "patient_id" is available.

**Limited implementation of structured summaries.** We find that overall HuggingFace follows a much more structured and complete documentation than Kaggle, as reflected in their guides [1, 3]. From our list of MI datasets, on HF we find an instance of MIMIC-CXR with no documentation or reference at all, and other medical datasets (*e.g.* Alzheimer's disease or skin cancer classification) without source citation. We find the lack of source citations for patient-related data deeply concerning. Kaggle automatically computes *usability score*, which is associated with the tag "well-documented" and used for ranking results when searching for a dataset. This score is based on completeness, credibility, and compatibility, we show detailed information about these parameters in Section A.1 of the Supplementary Material. However, we find that even datasets with 100% usability present some issues. For example, based on our observations, the parameter *update frequency* from maintenance is rarely used. However, an option for this parameter is to set it as "never" while still achieving a high *usability score*. Details about *provenance* might be filled in on the data card but may be vague, such as "uses internet sources".

We compare the categories analyzed in Kaggle and HuggingFace's data cards with those in Datasheets [40]. Despite making various efforts to integrate dataset documentation, such as the recent inclusion of Croissant [5], a metadata format designed for ML-ready datasets, we have noticed a prevalent issue:

many of the documentation fields remain empty. While these platforms strive to provide structured summaries, the practical outcome often falls short. Overall, we find composition and collection process are the two fields most represented; motivation of the creation of the dataset is rarely included in the general description of the dataset; information about preprocessing/cleaning/labeling or about uses is usually missing. Only for HuggingFace the field *task_categories* can point to some possible uses, potentially enabling systematic analysis of a specific task or tag. Kaggle provides a parameter for maintenance of the dataset, although we have already mentioned its limitations. HuggingFace does not provide a specific parameter for maintenance but it is possible to track on their website the history of files and versions. We detail the parameter categorization in Table 2 (Suppl. Material).

## 4 Discussion

**Asymmetry between open data and proprietary datasets.** Commercial AI systems in clinical settings are unlikely to rely solely on open MI datasets for training. They ensure data quality through agreements or obtaining high-quality medical images [95]. Companies providing proprietary MI datasets or labeling services handle challenges such as licensing, documentation, and data quality, offering greater customization and flexibility. Such proprietary datasets remain unaffected by the mentioned challenges [95, 130]. Similarly, [130] have shown how regulatory compliance and internal organizational requirements, transverse and often define dataset quality.

This asymmetry between the issues of open data and the value offered by proprietary datasets highlights the shortcomings of publicly available MI data. While open data initiatives like CCPs offer the potential to redistribute data value for the common good and public interest, the current status of MI datasets falls short in reliably training high-performing, equitable, and responsible AI models. Due to these limitations, we suggest rethinking and evaluating open datasets within CCPs through the concepts of *access, quality, and documentation* drawing upon the FAIR principles [122]. We argue that these concerns need to be accounted for if the MI datasets are to live up to the ideals of open data.

**Access to open datasets should be predictable, compliant with open licensing, and persistent.** In this paper, we show that a proper dataset infrastructure (both legal and technical) is crucial for their effective utilization. Open datasets must be properly licensed to prevent harm to end-users by models trained on legally ambiguous open data with the potential for bias and unfairness [104, 69]. Moreover, vague licensing pushes the users of open datasets into a legal grey zone [41]. [28] noticed such a legal gap in the "inappropriate" use of open AI models and pointed out the danger of their possible unrestricted and unethical use. To ensure the responsible use of AI models, they envisioned enforceable licensing. Legal clarity should also span persistent and deterministic storage. The most popular ML datasets are mostly hosted by established academic institutions. However, the CCPs host a plethora of duplicated or altered MI datasets. Instead of boosting the opportunities for AI creators, this abundance may become a hindrance when *e.g.*, developers cannot possibly track changes introduced between different versions of a dataset. We argue that open data has to be predictably and persistently accessible under clear conditions and for clear purposes.

**Open datasets should be evaluated against the context of real-world use.** The understanding of high-quality data for AI training purposes is constantly evolving [120]. After a thorough evaluation focused on real-world use, MI datasets, once considered high-quality [52, 118, 22, 113], were revealed to contain flaws (chest drains, dark corners, ruler markers, etc.) questioning their clinical usefulness [86, 57, 15, 115]. Maintaining open datasets is often an endeavor that is too costly for their creators, resulting in the deteriorating quality of available datasets. Moreover, we showed the prevalence of information about shortcuts and missing metadata in MI datasets hosted on CCPs. These issues can reduce the clinical usefulness of developed systems and, in extreme scenarios, potentially cause harm to the intended beneficiaries. We encourage the MI and other ML communities to expand the understanding of high-quality data by incorporating rich metadata and emphasizing real-world evaluations, including testing to uncover biases or shortcuts [86, 43].

**Datasets documentation should be complete and up-to-date.** Research has shown that access to large amounts of data does not necessarily warrant the creation of responsible and equitable AI models [94]. Instead, it is the connection between the dataset's size and the understanding of the work that resulted in the creation of a dataset. This connection is the premise behind the creation of proprietary datasets designed for use in private enterprises. When that direct connection

is broken, a fairly common scenario in the case of open datasets, the knowledge of the decisions taken during dataset creation is lost. Critical data and data science scholars are concerned about the social and technical consequences of using such undocumented data. Thus, a range of documentation frameworks were proposed [48, 12, 94, 40, 34, 51]. Each documentation method slightly differs, focusing on various aspects of dataset quality. However, their overall goal is to introduce greater transparency and accountability in design choices. These conscious approaches aim to foster greater reproducibility and contribute to the development of responsible AI. Unfortunately, as shown in this paper, the real-world implementation of these frameworks is lacking. Even when a CCP provides a documentation framework, the content rarely aligns with the frameworks' principles. CCPs could take inspiration from PhysioNet [44], which implements checks on new contributions. Any new submissions are first vetted[7] by the editors and may require re-submission if the expected metadata is not provided. When the supplied documentation does not adhere to the frameworks' principles, it fails to fulfill its intended purpose, placing users of open datasets at a disadvantage compared to users of proprietary datasets. We note that while we talk about completeness of documentation and the frameworks provide guidelines on what kind of information that might entail, it is not clear how one would quantify that the documentation is 86% complete in a way that reflects the data stakeholders' needs and is not merely a box-ticking exercise.

**CCPs could benefit from commons-based governance.**  Data governance can help mitigate the issues of accountability, fairness, discrimination, and trust. Inspired by the Wikipedia model [37], we recommend that CCPs implement norms and principles derived from this commons-based governance model. We suggest incorporating at least the roles of *data administrator*, and *data steward*. We define the role of *data administrator* as the first-level of data stewardship, a sanctioning mechanism that ensures proper (1) licensing, (2) persistent identifiers, and (3) completeness of metadata for open MI datasets that enter the platform. We define as the second-level of data stewardship, the role of *data steward*, who will be responsible for the ongoing monitoring of the (1) maintenance, (2) storage, and (3) implementation of documentation practices.

Nevertheless, these data stewardship proposals, as a commons-based governance model, need further exploration within a broader community of CCP practitioners. Recognizing the limited resources (monetary and/or human labor) in CCP initiatives, we are very careful in suggesting a complex governance system that would solely rely on the unpaid labor of dataset creators. Instead, we propose this direction for future applied research to enhance the dataset management and stewardship of MI datasets on CCP through commons-based approaches. We sincerely hope that more institutions will support efforts to improve the value of open datasets, which will require additional structural support, such as permanent and paid roles for data stewards [90].

**Initiatives to work on data and improve the data lifecycle.**  Several fairly recent initiatives aim to address the overlooked role of datasets like the NeurIPS Datasets and Benchmarks Track or the Journal of Data-centric Machine Learning Research (DMLR). New develop platforms, like the data providence explorer [76], help developers track and filter thousands of datasets for legal and ethical issues, and allow scholars and journalists to examine the composition and origins of popular AI datasets. Other newly born initiative is Croissant [5], a metadata format for ML-ready datasets, which is currently supported by Kaggle, HuggingFace and other platforms. ML and NLP conferences have started to require ethics statements and various checklists with submissions [14, 18, 98] for the reviewer use, and even include them in the camera-ready versions of accepted papers [99] to incentive better documentation. Such checklists typically include questions about data license and documentation, and they could be extended to help develop the norm of not just sharing, but also documenting any new data accompanying research papers, or encourage the use of the 'official' documented dataset versions.

In the MI context, conferences like MICCAI have incorporated a structured format for challenge datasets to ensure high-quality data. Initiatives like Project MONAI [17] introduce a platform to facilitate collaborative frameworks for medical image analysis and accelerate research and clinical collaboration. Drawing inspiration from CV, benchmark datasets are now emerging in MI, such as MedMNIST [127] and MedMNIST v2 [128]. These multi-dataset benchmarks have their pros and cons. They are hosted on Zenodo, which facilitates version control, provides persistent identifiers, and ensures proper storage. However, the process of standardizing MI datasets to the CV format

---

[7]https://physionet.org/about/publish/

means they lack details about patient demographics (such as age, gender, and race), information on the medical acquisition devices used, and other metadata, including patient splits for training and testing. Recent works have investigated data sharing and citations practices at MICCAI and MIDL [109], and reproducibility and quality of MIDL public repositories [107].

**More insights needed from all people involved.**   A limitation of our study is that it is primarily based on our quantitative evidence and our subjective perceptions of the fields and practices we describe of a limited number of screened datasets, yet the most cited ones. However, a recent study [129] has quantitatively and qualitatively confirmed our observations about the lack of documentation for datasets on HuggingFace. However, we did not reach out to Kaggle or HuggingFace. To gain a better understanding of data curation, maintenance, and re-use practices, it would be valuable to do a qualitative analysis with MI and ML practitioners to understand their use of datasets. For example, [130] is a recent study, based on interviews with researchers from companies and the public health sector, of how several medical datasets were created. It would be interesting to investigate how researchers select datasets to work on, looking beyond mere correlations with popularity and quantitative metrics. We might be able to learn valuable lessons from other communities that we have not explored in this paper, for example neuroimaging (which might appear to be a subset of medical imaging, but in terms of people and conferences is a fairly distinct community), where various issues around open data have been explored [92, 91, 114, 121, 50, 11, 84].

However, we should not forget that understanding research practices around datasets is not just of relevance to ML and adjacent communities. These datasets have broader importance as these datasets are affecting people who are not necessarily represented at research conferences, so further research should involve these most affected groups [112]. Public participation in data use [42], alternative data sharing, documenting, and governance models [35] are crucial to addressing power imbalances and enhancing data's generation of value as a common good [87, 93, 111]. Furthermore, neglecting the importance of recognizing and prioritizing the foundational role of data when working with MI datasets can lead to downstream harmful effects, such as *data cascades* [102]. **In conclusion**, our observations reveal that the existing CCP governance model falls short of maintaining the necessary quality standards and recommended practices for sharing, documenting, and evaluating open MI datasets. Our recommendations aim to promote better data governance in the context of MI datasets to mitigate these risks and uphold the reliability and fairness of AI models in healthcare.

## Acknowledgments

This project has received funding from the Independent Research Council Denmark (DFF) Inge Lehmann 1134-00017B. We would like to thank the reviewers for their their valuable feedback, which has contributed to the improvement of this work.

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

# A   Supplementary Material

## A.1   Data Cards

Table 2 shows the extracted documentation parameters from Kaggle and HuggingFace, which we categorized according to Datasheets [40].

On **HuggingFace**, we find information about the annotation creators (*e.g.*, crowdsource, experts, ml-generated) or specific task categories (*e.g.*, image-classification, image-to-text, text-to-image). Such parameters can be used to filter results when searching on HuggingFace, potentially enabling systematic analysis of a specific task or tag.

On **Kaggle**, we notice that some important parameters shown in the dataset website such as *temporal and geospatial coverage*, *data collection methodology*, *provenance*, *DOI citation*, and *update frequency* cannot be automatically extracted with their API, so we manually included them.

Kaggle automatically computes a *usability score*, which is associated with the tag "well-documented", and used for ranking results when searching for a dataset. Kaggle's *usability score* is based on:

- Completeness: *subtitle*, *tag*, *description*, *cover image*.
- Credibility: *provenance*, *public noteboook*, *update frequency*.
- Compatibility: *license*, *file format*, *file description*, *column description*.

The *usability score* is based on only 4 out of 7 aspects from Datasheets [40].

| | Kaggle | HuggingFace |
|---|---|---|
| Motivation | *username*
*dataset name*
*description* | *username*
*dataset name*
*description* |
| Composition | *temporal coverage*
*geospatial coverage* | *size categories*: $n < 1K$, $1K < n < 10k$, $1M < n < 10M$
*language*: en, es, hi, ar, ja, zh, ...
*dataset info*: {image, class_label: bird, cat, deer, frog, ...}
*data splits*: training, validation
*region*
*version* |
| Collection | *data collection method*
*provenance* | *source dataset*: wikipedia, ...
*annotation creators*: crowdsourced, found, expert-generated, machine-generated, ... |
| Preprocessing cleaning / labeling | | |
| Uses | | *task_categories*: image-classification, image-to-text, question-answering
*task_ids*: multi-class-image-classification, extractive-qa, ... |
| Distribution | *license*: cc, gpl, open data commons, ...
*DOI citation* | *license*: apache-2.0, mit, openrail, cc, ... |
| Maintenance | *update frequency*: weekly, never, not specified, ... | |
| Other | *keywords*
*number of views*
*number of downloads*
*number of votes*
*usability rating* | *tags*
*number of likes*
*number of downloads in the last month*
*arXiv* |

Table 2: Documentation parameters extracted from Kaggle and HuggingFace categorized according to Datasheets [40], except the last rows (Other). We represent in italic the extracted parameter, and show examples values for them. We include *description* in Motivation, although we find that this parameter can contain any type of dataset information.

## A.2   Duplicates on Kaggle

We automatically retrieve all the duplicates for the top-10 listed MI datasets on Kaggle, as well as some popular datasets (suggested by the reviewers). In Table 3, we show the number of duplicates on Kaggle, the size of the original dataset, the cumulative size of the duplicates, and information about the license and description on Kaggle for the duplicates. We query the name of each dataset as shown in Table 3, except for DRIVE and NIH-CXR14. For NIH-CXR14, we use "nih chest x-ray" as query. When querying "DRIVE" (not case-sensitive) we got over 1800 datasets related to cars, Formula One, and similar topics. To refine results, we applied a case-sensitive filter, retaining only those with capitalized "DRIVE". We also queried Kaggle using "drive retina" and found 13 datasets, of which only 5 were new when compared to our filtered query. Combining the two set of results, we identified 41 duplicates.

| Dataset | Duplicates | Size | | License | | Description |
|---|---|---|---|---|---|---|
| | | Original | Kaggle | (%) | types | (%) |
| CheXpert [52] | 47 | 440.0 GB[*] | 342.1 GB | 19.1 | 4 | 10.6 |
| DRIVE [110] | 34 | 30.1 MB | **11.7 GB** | 26.5 | 5 | 85.3 |
| fastMRI [59] | 8 | 6.3 TB | 215.2 GB | 62.5 | 3 | 25.0 |
| LIDC-IDRI [8] | 43[†] | 69.0 GB | **539.7 GB** | 20.9 | 6 | 18.6 |
| NIH-CXR14 [118] | 47 | 42.0 GB | **654.6 GB** | 59.6 | 5 | 97.9 |
| HAM10000 [113] | 141 | 3.0 GB | **468.4 GB** | 42.6 | 11 | 26.9 |
| MIMIC-CXR [58] | 13 | 554.2 GB | 62.1 GB | 46.2 | 4 | 23.1 |
| Kvasir-SEG [56] | 51 | 66.9 MB | **8.7 GB** | 41.2 | 4 | 15.7 |
| STARE [49] | 10 | 504.4 MB | **11.9 GB** | 30.0 | 2 | 40.0 |
| LUNA [105] | 46 | 66.7 GB | **585.6 GB** | 19.6 | 3 | 10.9 |
| BraTS [80] | 383 | 51.5 GB[§] | **7.3 TB** | 30.0 | 9 | 92.4 |
| ACDC [13] | 28 | 2.3 GB | **127.7 GB** | 28.6 | 5 | 14.3 |
| ADNI [53] | 70 | N/A[¶] | 803.3 GB | 57.1 | 4 | 40.0 |
| OASIS [61, 66, 78, 79] | 53 | 34.5 GB[‡] | **657.7 GB** | 56.6 | 3 | 15.1 |

Table 3: Information of the medical imaging dataset duplicates on Kaggle: number of duplicates; size of the original dataset and the storage on Kaggle; license information of the duplicates, percentage reported and different types of licenses; percentage of descriptions from duplicates that contain any text. [*]CheXpert dataset is 440 GB, however the 11 GB subset is the most commonly used and reshared. [†]We do not count LUNA duplicates for LIDC-IDRI. [§]BraTS datasets originated from challenges (2012-2022). These datasets are hosted at different websites and we couldn't retrieve their total size, dataset size is estimated from BraTS 2023. [¶]The size details of the ADNI dataset were not readily available. We submitted an "ADNI Use Application" request but did not receive access in time. [†]OASIS dataset have 4 series, however, we only had access to the size information of OASIS-1 and OASIS-2, so the size estimation is based on these two series. We highlight in boldface when the cumulative size on Kaggle is larger than the original size. Data was collected in October, 2024.

We review each list and eliminate duplicates that are not relevant due to ambiguity, such as music datasets for OASIS. Some datasets were difficult to disambiguate because they contained no descriptions and provided compressed information (*e.g.*, npy files). We also found pretrained models listed under the dataset category. We decided to keep the examples we could not disambiguate and the pretrained models, as they were only a few. We keep duplicates that are aggregation of datasets, e.g. one instance groups together 3 different datasets for Alzheimer's, Parkinson's and "normal", which can cause data leakage [100]. LUNA was a challenge dataset created after LIDC-IDRI. We do not count LUNA-16 duplicates as duplicates of LIDC-IDRI, we only consider them for LUNA.

