# OpenReview forum: "Copycats: the many lives of a publicly available medical imaging dataset"
_NeurIPS.cc/2024/Datasets_and_Benchmarks_Track — NeurIPS 2024 Track Datasets and Benchmarks Poster_

### Official Review · Reviewer_XUa5 · 2024-07-21
**Good paper with some limitations on the number of MI datasets analyzed**

**Rating:** 8
**Confidence:** 5
**Correctness:** Yes.
**Clarity:** Yes

**Review:**

Pros:
+ This paper covers a very important element which is the fact that MI datasets need to be handled with care and cannot be treated in the same way as CV datasets.

+ The message is very relevant for this community as, many ML practitioners do not have the background on healthcare to understand the critical aspects of medical data. This paper aims to bring awareness on the topic.

+ The paper is clear, well written and easy to follow

Cons:
- For a paper focusing on MI datasets, one would expect a larger coverage. There are numerous very famous datasets that are out of the list. To cite a few: BraTs, Isles, ACDC, ADNI, OASIS, among others
- Also, it would have been important to discuss less famous datasets. It is very likely that those are less rigourous in the maintenance. It is important to point to that.

**Strengths:**

As written in the review this paper covers a very important element which is the fact that MI datasets need to be handled with care and cannot be treated in the same way as CV datasets. The message is very relevant for this community as, many ML practitioners do not have the background on healthcare to understand the critical aspects of medical data. This paper aims to bring awareness on the topic.

**Additional Feedback:**

This is a nice work. It would be great to see it expanded by including more MI datasets.

**Documentation:**

Yes

**Limitations:**

No. The authors could discuss the limitations that this analysis has regarding the limited size of MI datasets under study.

**Opportunities For Improvement:**

While the comparison to other non-MI datasets is important, the conclusions drawn from there are not really interesting for the message the paper aims to transmit (that they use more regulated platforms? It would have been better to see a larger set of MI datasets analysed. That will send a stronger message

**Relation To Prior Work:**

There is no related work. This work is first of its kind. There is a background section providing the necessary elements.

**Summary And Contributions:**

This paper presents an overview of publicly available medical imaging (MI) datasets. The point that the paper aims to bring forward is that MI datasets cannot be treated in a similar way as computer vision (CV) papers, due the sensitive nature of healthcare-related data.

The paper:
- Provides an overview of the key differences between MI and CV datasets.
- Presents elements of data governance that are relevant for healthcare + machine learning
- Goes through the documentation of 30 well known datasets to identify issues associated to vague licenses, lack of persistent links, etc.

---

> ### Author Rebuttal · Authors · 2024-08-16
>
> We would like to thank Reviewer XUa5 for their feedback.
>
> The reviewer found our message very relevant for this community as, many ML practitioners do not have the background on healthcare to understand the critical aspects of medical data. They also highlighted that "There is no related work. This work is first of its kind."
>
> We address their concerns below.
>
> > For a paper focusing on MI datasets, one would expect a larger coverage. There are numerous very famous datasets that are out of the list. To cite a few: BraTs, Isles, ACDC, ADNI, OASIS, among others
>
> When we obtained the list of datasets, we were also very surprised that the datasets the reviewer mentioned, are not there - to us they feel more widely used. But we opted to select datasets that are often used in a more systematic way, which PapersWithCode provided. A recent study [11] showed that citations of datasets do not often reflect their use and vice versa (and that study does in fact include some of the datasets the reviewer mentions). Therefore we opted for PapersWithCode since this is more likely to reflect actual use.
>
> We do agree it is important especially for readers outside medical imaging to know understand the distinction between the selected datasets so we added a few lines explaining this point. In "Study setup" paragraph in Section 3. Findings, we added "We were expecting to find MI datasets like BraTs [1], ACDC [15], etc. in the list, since we thought they are commonly used, but we decided to leverage Papers with Code to retrieve datasets in a systematic way. "
>
> Furthermore, we also added BRATS to our analysis of and we will aim to add more MI datasets before the discussion ends. We added "On Kaggle, we found 350 datasets related to BraTS (Brain Tumor Segmentation) [1]. BraTS datasets originated from challenges (2012-2022). These datasets are hosted at different websites and we couldn't retrieve their total size to run a similar comparison to ISIC."
>
> > Also, it would have been important to discuss less famous datasets. It is very likely that those are less rigorous in the maintenance. It is important to point to that
>
> We disagree that dataset popularity is correlated to dataset maintenance, there exist counterexamples in both directions. We provide some examples of famous datasets with poor practices in our study in lines 134-136 "For instance, the LIDC-IDRI website uses red font size to signal errors and updated labels, while the NIH-CXR14 website hosts derivative datasets and job announcements.". Unfortunately, less famous datasets are by definition more difficult to investigate systematically. Less famous but well-documented datasets, like PAD-UFES-20, may originate from lesser-known institutions and receive less attention, for example due to the prestige of the institution.
>
> > While the comparison to other non-MI datasets is important, the conclusions drawn from there are not really interesting for the message the paper aims to transmit (that they use more regulated platforms? It would have been better to see a larger set of MI datasets analysed. That will send a stronger message
>
> We apologize for missing the link between non-MI datasets and our conclusions. We have included the justification in the "Study setup", and highlighted the shared (with CV and NLP) and unique issues for MI datasets on CCPs. We have also rearranged the presentation of Section 3. Findings to make the paper more comprehensive.
> * In "Study setup", we added "We aim to promote better practices in the context of MI datasets. We include in the comparison CV and NLP because MI is often inspired by these other ML areas, and is where data governance have received more attention recently."
> * We have rearranged the order of the paragraphs to present first "Lack of persistent identifiers and storage" and we have merged this paragraph with "Missing licenses on CCPs".
> * In "Duplicate datasets and missing metadata on CCPs", we rephrased and added "The lack of documentation for all ML datasets hampers tracking their usage, potentially violates sharing agreements or licenses, and hinders reproducibility. Moreover, due to the characteristics of MI datasets, models trained on datasets missing metadata could result into overoptimistic performance due to data splits mixing patient data, or bias [2] or shortcuts [3,4]."
>
> References
>
> [1] Menze, B. H., Jakab, A., Bauer, S., Kalpathy-Cramer, J., Farahani, K., Kirby, J., ... & Van Leemput, K. (2014). The multimodal brain tumor image segmentation benchmark (BRATS). IEEE transactions on medical imaging, 34(10), 1993-2024.
>
> [2] Larrazabal, A. J., ... & Ferrante, E. (2020). Gender imbalance in medical imaging datasets produces biased classifiers for computer-aided diagnosis. Proceedings of the National Academy of Sciences, 117(23), 12592-12594.
>
> [3] Oakden-Rayner, L., ... & Ré, C. (2020, April). Hidden stratification causes clinically meaningful failures in machine learning for medical imaging. In Proceedings of the ACM conference on health, inference, and learning (pp. 151-159).
>
> [4] Winkler, J. K., ... & Haenssle, H. A. (2019). Association between surgical skin markings in dermoscopic images and diagnostic performance of a deep learning convolutional neural network for melanoma recognition. JAMA dermatology, 155(10), 1135-1141.
>
> [11] Sourget, T., ... & Cheplygina, V. (2024). [Citation needed] Data usage and citation practices in medical imaging conferences. arXiv preprint arXiv:2402.03003.
>
> [15] Bernard, O., Lalande, A., Zotti, C., Cervenansky, F., Yang, X., Heng, P. A., ... & Jodoin, P. M. (2018). Deep learning techniques for automatic MRI cardiac multi-structures segmentation and diagnosis: is the problem solved?. IEEE transactions on medical imaging, 37(11), 2514-2525.

---

> > ### Author Rebuttal · Authors · 2024-08-16
> >
> > (continuation of rebuttal)
> >
> > > This is a nice work. It would be great to see it expanded by including more MI datasets.
> >
> > We thank the reviewer for the appreciation of our work. We have clarified the systematic way in which we selected the datasets from Papers with code as mentioned above. We have also included one more dataset to our duplication investigation (BraTS) and will try to add some more.
> >
> > We also have included multi-datasets benchmarks MedMNIST and MedMNIST v2 to our Section 4. Discussion.
> >
> > We added "Drawing inspiration from CV, benchmark datasets are now emerging in MI, such as MedMNIST [9] and MedMNIST v2 [10]. These multi-dataset benchmarks have their pros and cons. They are hosted on Zenodo, which facilitates version control, provides persistent identifiers, and ensures proper storage. However, the process of standardizing MI datasets to the CV format means they lack details about patient demographics (such as age, gender, and race), information on the medical acquisition devices used, and other metadata, including patient splits for training and testing."
> >
> > References
> >
> > [9] Yang, J., ... & Ni, B. (2021, April). Medmnist classification decathlon: A lightweight automl benchmark for medical image analysis. In 2021 IEEE 18th International Symposium on Biomedical Imaging (ISBI) (pp. 191-195). IEEE.
> >
> > [10] Yang, J., ... & Ni, B. (2023). Medmnist v2-a large-scale lightweight benchmark for 2d and 3d biomedical image classification. Scientific Data, 10(1), 41.

---

> > > ### Comment · Reviewer_XUa5 · 2024-08-19
> > >
> > > Thanks for the detailed response.
> > >
> > > As per my initial review, I think this work is of high importance to the ML (and CV) communities that often lack sufficient background on MI and why the datasets used there cannot be handled in the same way as the ones typically used in ML/CV.
> > >
> > > Some of my concerns have been clarified. I have updated my score to show my support for this work.

---

> > > > ### Author Response · Authors · 2024-08-20
> > > >
> > > > We would like to once again thank the reviewer for their comments, for recognizing the importance of our work and for updating the score to reflect their support. We sincerely appreciate the time and effort invested in reviewing and providing this valuable feedback.

---

### Official Review · Reviewer_Lyfj · 2024-07-24
**Good summary of issues in medical image datasets**

**Rating:** 6
**Confidence:** 4
**Clarity:** The paper is clear and well-written.

**Review:**

The paper touches upon an important issue for the ML community and the potential for discussion would be impactful. The paper correctly identifies unique aspects of medical imaging data, data principles (FAIR), and societal issues such as data governance. However, some points are not specific to medical image datasets such as data duplication, and would benefit from comparison to vision datasets to see if the issue is relatively worse in the medical domain. While the work identifies major issues with current data practices, it is unclear whether the provided recommendations are specific enough to remedy these issues. Additionally, the recommendations could just as easily be applied to non-medical data. Overall, the paper is well-written and has good potential for impactful discussion for the DMLR community.

**Strengths:**

The papers address an important and relevant topic for future research of reproducibility, dataset quality, and data management for medical imaging datasets. More researchers in the ML community should consider the data challenges and recommendations discussed in the paper. The emphasis on data governance and stewardship is especially important for medical data and the example of physionet is a managed platform with documentation and quality standards. Now the question is how to emulate and incentivize better data management practices.

**Additional Feedback:**

* Add MedMNIST datasets to case study
* In line 82, say what the effects are
* Look into the Data providence initiative https://www.dataprovenance.org/explorer-introduction
* Paragraph 124-129 seems out of place and better fit to the previous section
* In line 165, how to enforce better documentation? What are your recommendations for change?
* Breifly explain what TRIPOD and CONSORT guidelines are and how they help for medical data.
* Explain why regulated platforms are important for medical imaging data and why TF and PyTorch should be given consideration instead of physionet. As general DL frameworks, it makes sense that they would not include medical datasets. Indeed they often don't support the domain specific operations such as 3D operations of MONAI, which is made specifically for medical imaging deep learning.
* Add more datasets to the case study. If NLP text datasets are included, medical text datasets such also be included for comparison.
* It would be worth looking at conferences such as https://www.midl.io and MICCAI to see current practices for data management and tracking. How could insights from more targeted communities be translated to the broader ML audience
* Lines 242-243 are unclear and could be reworded
* HOw to fix missing documentation or licensing violations on platforms like Kaggle? Reporting systems? Or something else?
* In line 293, cite positive examples for references
* In line 287, give examples of flaws
* How to enhance transparency and documentation in cost-effective, low-friction ways? Could automation, templates, or LLM be useful here?
* How to create incentives for producing high-quality open datasets. Datasets track is a good start but what are other ways? How to reward the continuous improvement and fixing of existing datasets?

**Correctness:**

The claims are mostly supported with some caveats such as the study of popular datasets could be expanded such as to more datasets and medical domains to show that this is indeed a systematic issue. The recommendations could be supported with surveys or other methods to back up that the recommendations are practical and realistic to be implemented by researchers and platforms.

**Documentation:**

N/A

**Ethics:**

No ethical concerns.

**Limitations:**

Some of the issues brought up in the paper are not entirely specific the medical imaging datasets and there is a lack of comparison showing that these issues were worse than in comparable domains such as computer vision. For instance, line 130 mentions that practices for dataset updates and tracking have not been formalized for MI but could also be argued that they have not been properly formalized for CV either. Additionally, data duplication is a major issue for CV as well. A more fair comparison would be needed to state that the issue is comparatively worse in MI. It is unclear that the case study in Table 1 shows this as more MI datasets are licenses than CV datasets. It is also unclear why RP such as TF and Keras should be compared for MI datasets. A more fair comparison would be using medical domain frameworks such as MONAI which provides support for 3D datasets and transformations, which PyTorch and Keras do not support. Many issues such as documentation and dataset quality are not unique to medical imaging datasets, and it is unclear that medical imaging faces worse practices in general than common CV or ML datasets.

**Opportunities For Improvement:**

Explain why being on a generic RP such as TF or Keras is important for medical datasets rather than a medical-specific RP such as MONAI. A discussion of 3D/4D in MI and CV is entirely missing, as well as considerations of anatomical/physiological/physical correctness. These would be more MI-specific than documentation or data duplication issues.

**Relation To Prior Work:**

More discussion of efforts to improve medical datasets benchmarks such as MedMNIST (https://www.nature.com/articles/s41597-022-01721-8) could be discussed.

**Summary And Contributions:**

The paper summaries many open challenges of public medical datasets such as anonymization, licensing, and documentation. They conduct a case study analysis of popular vision, text, and medical datasets on open platforms such as Huggingface and Kaggle. Some recommendations for better data practices are provided.

---

> ### Author Rebuttal · Authors · 2024-08-16
>
> We thank reviewer Lyfj for their time and valuable feedback.
>
> The reviewer recognized the topic of our paper as an important issue for the ML community and considered the potential for discussion would be impactful. The reviewer stated how our paper correctly identifies unique aspects of medical imaging data, data principles (FAIR), and societal issues such as data governance. Additionally, the reviewer highlighted the importance and relevance of this topic for future research of reproducibility, dataset quality, and data management for medical imaging datasets. More researchers in the ML community should consider the data challenges and recommendations discussed in the paper. We agree with the reviewer that the emphasis on data governance and stewardship is especially important for medical data.
>
> We understand the concerns of the reviewer. We clarify them below.
>
> >  Explain why being on a generic RP such as TF or Keras is important for medical datasets rather than a medical-specific RP such as MONAI.
>
> We apologize for the confusion. We have clarified in Section 3. Findings our motivation to compare MI with CV and NLP. For that reason we picked TF and Keras to compare where datasets live across different platforms.
>
> We agree that medical-specific platforms such as MONAI can help CV and MI researchers, so we have included a discussion of project MONAI in "Initiatives to work on data and improve the data lifecycle", we added "Initiatives like Project MONAI [5] introduce a platform to facilitate collaborative frameworks for medical image analysis and accelerate research and clinical collaboration."
>
> We understand your comment this way, but if we haven't fully addressed your concern, please share your thoughts or suggestions with us.
>
> > A discussion of 3D/4D in MI and CV is entirely missing, as well as considerations of anatomical/physiological/physical correctness
>
> We acknowledge the limitation about our study.
>
> In the paragraph "Not just “small computer vision”!" of Section 2.1 we mentioned how splitting 3D scans could be problematic if the obtained 2D slices are just treated as independent images. We agree that considerations of anatomical/physiological/physical correctness are very important. In this same paragraph, we have added "3D images from modalities like MRI can vary significantly depending on the sequence used. For example, brain MRI sequences (T1-weighted, T2, FLAIR, etc.) are designed to emphasize different brain structures, offering specific physiological and anatomical details".
>
> We thank the reviewer for suggesting this improvement. We hope to have addressed this concern, but if our response isn’t satisfactory, please share your comments so we can discuss further changes.
>
> > A more fair comparison would be using medical domain frameworks such as MONAI which provides support for 3D datasets and transformations, which PyTorch and Keras do not support.
>
> We appreciate the reviewer reminding us about MONAI. We did not include MONAI in our case study or in Table 1 because we found too few datasets. We found: MedNISTDataset, DecathlonDataset, TciaDataset. These datasets are by design aggregates of multiple datasets. We believe that this would be an important topic to explore in the future. We have included MONAI project in the paragraph "Initiatives to work on data and improve the data lifecycle" of Section 4. In particular, we added "Initiatives like Project MONAI [5] introduce a platform to facilitate collaborative frameworks for medical image analysis and accelerate research and clinical collaboration."
>
> References
>
> [5] Cardoso, M. J., Li, ... & Feng, A. (2022). MONAI: An open-source framework for deep learning in healthcare. arXiv preprint arXiv:2211.02701.

---

> > ### Author Rebuttal · Authors · 2024-08-16
> >
> > (continuation of rebuttal - 2/4)
> >
> > > For instance, line 130 mentions that practices for dataset updates and tracking have not been formalized for MI but could also be argued that they have not been properly formalized for CV either. Additionally, data duplication is a major issue for CV as well. A more fair comparison would be needed to state that the issue is comparatively worse in MI.
> > > Many issues such as documentation and dataset quality are not unique to medical imaging datasets, and it is unclear that medical imaging faces worse practices in general than common CV or ML datasets.
> >
> > We agree that dataset management practices issues (such as data duplication or reproducibility of results) are common to ML datasets. However, we believe that CV and NLP data management practices have received more attention [6,7,8]. Moreover, we wanted to highlight how the characteristics of MI datasets (demographics, images belonging to a single patient, etc), as explained in Section 2.1, present some unique problems for MI datasets.
> >
> > We have restructured Section 3 to present the findings in a clearer way and highlighted the shared issues (with CV and NLP) and the unique ones for MI datasets on CCPs.
> > * In "Study setup", we added "We aim to promote better practices in the context of MI datasets. We include in the comparison CV and NLP because MI is often inspired by these other ML areas, and is where data governance have received more attention recently."
> > * We have rearranged the order of the paragraphs to present first "Lack of persistent identifiers and storage" and we have merged this paragraph with "Missing licenses on CCPs".
> > * In "Duplicate datasets and missing metadata on CCPs", we rephrased and added "The lack of documentation for all ML datasets hampers tracking their usage, potentially violates sharing agreements or licenses, and hinders reproducibility. Moreover, due to the characteristics of MI datasets, models trained on datasets missing metadata could result into overoptimistic performance due to data splits mixing patient data, or bias [2] or shortcuts [3,4]."
> >
> > > More discussion of efforts to improve medical datasets benchmarks such as MedMNIST could be discussed.
> >
> > We agree with the reviewer and have extended our discussion including more efforts to improve medical datasets, such as project MONAI, multi-dataset benchmarks and recent works on MI about data citation practices and reproducibility.
> >
> > We added "Initiatives like Project MONAI [5] introduce a platform to facilitate collaborative frameworks for medical image analysis and accelerate research and clinical collaboration. Drawing inspiration from CV, benchmark datasets are now emerging in MI, such as MedMNIST [9] and MedMNIST v2 [10]. These multi-dataset benchmarks have their pros and cons. They are hosted on Zenodo, which facilitates version control, provides persistent identifiers, and ensures proper storage. However, the process of standardizing MI datasets to the CV format means they lack details about patient demographics (such as age, gender, and race), information on the medical acquisition devices used, and other metadata, including patient splits for training and testing. Recent works have investigated data sharing and citations practices at MICCAI and MIDL [11], and reproducibility and quality of MIDL public repositories [12]."
> >
> > > Line 82: add what are the harmful effects for data cascades
> >
> > We added "*Data cascades* can lead to degraded model performance, reinforce biases, increase maintenance costs, and reduce trust in AI systems.  These problems often stem from poor data quality, lack of domain expertise, and insufficient documentation, which become increasingly difficult to correct once models are deployed."
> >
> > > Look into the Data providence initiative
> >
> > We have integrated the Data providence explorer in "Initiatives to work on data and improve the data lifecycle" of Section 4. Discussion.
> >
> > We added "New develop platforms, like the data providence explorer [13], help developers track and filter thousands of datasets for legal and ethical issues, and allow scholars and journalists to examine the composition and origins of popular AI datasets."
> >
> > > Paragraph 124-129 seems out of place and better fit to the previous section
> >
> > We presented this paragraph in Section 2.1 because errors in annotations or data, or shortcuts need to be reported and updated.
> >
> > References
> >
> > [6] Gebru, T., ... & Crawford, K. (2021). Datasheets for datasets. Communications of the ACM, 64(12), 86-92.
> >
> > [7] Hutchinson, B., ... & Mitchell, M. (2021, March). Towards accountability for machine learning datasets: Practices from software engineering and infrastructure. In Proceedings of the 2021 ACM Conference on Fairness, Accountability, and Transparency (pp. 560-575).
> >
> > [8] Bender, E. M., & Friedman, B. (2018). Data statements for natural language processing: Toward mitigating system bias and enabling better science. Transactions of the Association for Computational Linguistics, 6, 587-604.
> >
> > [9] Yang, J., ... & Ni, B. (2021, April). Medmnist classification decathlon: A lightweight automl benchmark for medical image analysis. In 2021 IEEE 18th International Symposium on Biomedical Imaging (ISBI) (pp. 191-195). IEEE.
> >
> > [10] Yang, J., ... & Ni, B. (2023). Medmnist v2-a large-scale lightweight benchmark for 2d and 3d biomedical image classification. Scientific Data, 10(1), 41.
> >
> > [11] Sourget, T., ... & Cheplygina, V. (2024). [Citation needed] Data usage and citation practices in medical imaging conferences. arXiv preprint arXiv:2402.03003.
> >
> > [12] Simkó, A., ... & Löfstedt, T. (2024, January). Reproducibility of the Methods in Medical Imaging with Deep Learning. In Medical Imaging with Deep Learning (pp. 95-106). PMLR.
> >
> > [13] Longpre, S., ... & Hooker, S. (2023). The data provenance initiative: A large scale audit of dataset licensing & attribution in ai. arXiv preprint arXiv:2310.16787.

---

> > ### Author Rebuttal · Authors · 2024-08-16
> >
> > (continuation of rebuttal - 3/4)
> >
> > > In line 165, how to enforce better documentation? What are your recommendations for change?
> >
> > We apologize for the misunderstanding. In line 165, we wanted to motivate that the current CCP data governance model fails to uphold the quality needed and recommended practices for sharing, documenting, and evaluating datasets. Our recommendations for better documentation and sharing practices are presented in the paragraph "CCPs could benefit from commons-based governance" of Section 4. Discussion. In particular, "we recommend that CCPs implement norms and principles derived from this commons-based governance model. We suggest incorporating at least the roles of data administrator, and data steward."
> >
> > > Briefly explain what TRIPOD and CONSORT guidelines are and how they help for medical data.
> >
> > We apologize for missing TRIPOD and CONSORT definitions. We have now included them in the manuscript.
> >
> > The Transparent Reporting of a multivariable prediction model for Individual Prognosis or Diagnosis (TRIPOD) guidelines provide statistical recommendations for the reporting of studies developing, validating, or updating a prediction model.
> >
> > The CONSORT guidelines are recommendations for reporting randomized controlled trials clearly and transparently. They include a checklist and flow diagram to ensure all critical aspects of study design, methodology, results, and interpretation are thoroughly reported
> >
> > TRIPOD and CONSORT guidelines are useful to take into account underrepresented populations and ensure transparency for ML applications.
> >
> > > Add more datasets to the case study. If NLP text datasets are included, medical text datasets such also be included for comparison.
> >
> > Exploring medical text datasets would definitely be interesting. Our current focus is on the most popular medical datasets (medical imaging). Future research could consider investigating multi-modal datasets, including text.
> >
> > > It would be worth looking at conferences such as https://www.midl.io and MICCAI to see current practices for data management and tracking. How could insights from more targeted communities be translated to the broader ML audience
> >
> > We agree that this is an interesting topic. [11] investigated dataset citation and sharing practices at MIDL and MICCAI conferences. This study found that fewer than half of the datasets analyzed were properly cited, which can hinder reproducibility and proper attribution.
> >
> > In "Initiatives to work on data and improve the data lifecycle" in Section 4. Discussion, we added "\added{Recent works have investigated data sharing and citations practices at MICCAI and MIDL [11], and reproducibility and quality of MIDL public repositories [12].}"
> >
> > > Lines 242-243 are unclear and could be reworded
> >
> > We have reworded lines 242-243 to be more comprehensive.
> >
> > "For example, the parameter *update frequency* from maintenance is rarely used. However, an option for this parameter is to set it as "never" while still achieving a high *usability score*. Details about *provenance* might be filled in on the data card but may be vague, such as 'uses internet sources'."
> >
> > > How to fix missing documentation or licensing violations on platforms like Kaggle? Reporting systems? Or something else?
> >
> > Inspired by commons-based governance, we suggest the administrator and the steward roles in Section 4. Huggingface currently supports the option to report a dataset.
> >
> > > In line 293, cite positive examples for references
> >
> > We thank the reviewer for the suggestion. We have added [3,14].
> >
> > >  In line 287, give examples of flaws
> >
> > We have included examples of flaws in line 287. For instance, in chest X-rays, chest drains may introduce bias when detecting pneumothorax. Similarly, in skin lesion classification, artifacts like dark corners, rulers, or patch markers can skew the results. We added "(chest drains, dark corners, ruler markers, etc.)"
> >
> > > How to enhance transparency and documentation in cost-effective, low-friction ways? Could automation, templates, or LLM be useful here?
> >
> > As described in paragraph "Data governance, documentation, and data hosting practices" in Section 2.2, we believe that tools such as datasheets for datasets [6] can significantly improve transparency and provide more comprehensive documentation.
> >
> > We considered that the community might suggest LLMs to improve the transparency and documentation of datasets. As an anecdotal example we tried to see if ChatGPT could retrieve information about demographic variables in the CheXpert dataset (the dataset has gender and age variables upon download). However, this is not fully documented in the dataset´s description. When we asked ChatGPT, the LLM provided conflicting answers regarding the presence of gender attributes in the dataset.
> >
> > It would be interesting to systematically investigate the use of LLMs for dataset documentation in future research.
> >
> > References
> >
> > [3] Oakden-Rayner, L., Dunnmon, J., Carneiro, G., & Ré, C. (2020, April). Hidden stratification causes clinically meaningful failures in machine learning for medical imaging. In Proceedings of the ACM conference on health, inference, and learning (pp. 151-159).
> >
> > [6] Gebru, T., ... & Crawford, K. (2021). Datasheets for datasets. Communications of the ACM, 64(12), 86-92.
> >
> > [11] Sourget, T., ... & Cheplygina, V. (2024). [Citation needed] Data usage and citation practices in medical imaging conferences. arXiv preprint arXiv:2402.03003.
> >
> > [12] Simkó, A., ... & Löfstedt, T. (2024, January). Reproducibility of the Methods in Medical Imaging with Deep Learning. In Medical Imaging with Deep Learning (pp. 95-106). PMLR.
> >
> > [14] Gichoya, J. W., Banerjee, I., Bhimireddy, A. R., Burns, J. L., Celi, L. A., Chen, L. C., ... & Zhang, H. (2022). AI recognition of patient race in medical imaging: a modelling study. The Lancet Digital Health, 4(6), e406-e414.

---

> > ### Author Rebuttal · Authors · 2024-08-16
> >
> > (continuation of rebuttal - 4/4)
> >
> > > How to create incentives for producing high-quality open datasets. Datasets track is a good start but what are other ways? How to reward the continuous improvement and fixing of existing datasets?
> >
> > This is an active topic we are interested. Aside from making mandatory statements about sharing data and code at conferences or journals, we have started working on a collaborative living review. We aim to document extended annotations, shortcuts, errors in annotations, derived datasets, etc. The incentive in this case would be to be a co-author of the manuscript.

---

> > > ### Comment · Reviewer_Lyfj · 2024-08-19
> > > **Still missing important areas of discussion**
> > >
> > > I thank the authors for their response. I still think the article is missing several important areas of discussion that would greatly benefit the paper. If these topics are addressed sufficiently, I will revise my rating.
> > >
> > > **Existing tools and standard for healthcare data**
> > > The paper does not mention or discuss existing healthcare data standards for ML such as the Fast Health Interoperability Resources (FHIR) that allows standardized data access REST architectures and JSON data formats [1] or specialized Python libraries for radiology/pathology images [2], or DL frameworks for medical imaging [3]; these are just a few examples, you should search for more. If the aim is to improve data quality and practices for MI in ML, then there should be more focus on the existing data standards and software already available but unknown to the broader ML community. Specifically, it would be helpful to point to examples from other papers to detail the challenges with working with clinical data and deploying clinical AI applications.
> > >
> > > **Better differentiating in unique challenges for MI data**
> > > The paper should better clarify the unique challenges of MI data. How to show that data duplication, missing data, or other quality issues are much worse in MI than CV in general? The findings of the study in Section 3 are not detailed enough to show this convincingly. Instead of only mentioning license and availability through CCP/RP, it should quantify the quality and documentation of the data. Additionally, the sample size of 10 MI and 10 CV datasets is too small. Otherwise, the arguments for the unique challenges of MI should focus on the challenges of interpreting and annotating MI data, the data dimensionality (3D/4D fMRI), or other domain-specific characteristics such as difficulty in generalization across institutions, etc. More subdomain-specific considerations could be considered such as the computational challenges in processing Whole-Slide Images (WSI) for pathology [6] or the complex preprocessing in radiology and neuroscience such as windowing, registration, and resampling --- along with the reproducibility challenges associated with these preprocessing steps. How to succinctly and effectively convey these challenges to ML practitioners? Providing example images of MI processing may be helpful here.
> > >
> > > **Social, ethical, and legal challenges of MI data**
> > > The paper should provide more discussion on the associated ethical and social challenges with working with medical data, starting from data acquisition and governance. Topics of interest include linkage attacks [7], privacy regulations such as HIPPA [8], standards for reporting secondary findings [9], the commodification of health data [10], alternative data governance structures [11], and public participation in data use [12]. More inclusion of these topics would further help distinguish MI challenges from CV data.
> > >
> > > - [1] Ayaz, Muhammad, et al. "The Fast Health Interoperability Resources (FHIR) standard: systematic literature review of implementations, applications, challenges and opportunities." JMIR medical informatics 9.7 (2021).
> > > - [2] Bridge, Christopher P., et al. "Highdicom: A python library for standardized encoding of image annotations and machine learning model outputs in pathology and radiology." Journal of digital imaging 35.6 (2022).
> > > - [3] Cardoso, M. Jorge, et al. "Monai: An open-source framework for deep learning in healthcare." arXiv preprint arXiv:2211.02701 (2022).
> > > - [4] Mincu, Diana, and Subhrajit Roy. "Developing robust benchmarks for driving forward AI innovation in healthcare." Nature Machine Intelligence 4.11 (2022).
> > > - [5] Lu, Charles, et al. "Deploying clinical machine learning? Consider the following..." arXiv preprint arXiv:2109.06919 (2021).
> > > - [6] Cui, Miao, and David Y. Zhang. "Artificial intelligence and computational pathology." Laboratory Investigation 101.4 (2021): 412-422.
> > > - [7] Sweeney, L. "Only you, your doctor, and many others may know. Technology Science (2015)." (2018).
> > > - [8] Mandl, Kenneth D., and Eric D. Perakslis. "HIPAA and the leak of “deidentified” EHR data." N Engl J Med 384.23 (2021): 2171-3.
> > > - [9] Weiner, Christine. "Anticipate and communicate: Ethical management of incidental and secondary findings in the clinical, research, and direct-to-consumer contexts" American Journal of Epidemiology 180.6 (2014).
> > > - [10] Hunter, Philip. "The big health data sale: as the trade of personal health and medical data expands, it becomes necessary to improve legal frameworks for protecting patient anonymity, handling consent and ensuring the quality of data." EMBO reports 17.8 (2016).
> > > - [11] Duncan, Jamie. "Data protection beyond data rights: Governing data production through collective intermediaries." Internet Policy Review 12.3 (2023).
> > > - [12] Ghafur, Saira, et al. "Public perceptions on data sharing: key insights from the UK and the USA." The Lancet Digital Health 2.9 (2020): e444-e446.

---

> > > > ### Author Response · Authors · 2024-08-20
> > > >
> > > > We would like to once again thank the reviewer for the interest in our work, the time and effort invested in reviewing, and for highlighting several areas for discussion. We will address these points in the coming days. However, due to space constraints, we may not be able to explore each point in detail. We will post again later this week.

---

> > > > ### Author Rebuttal · Authors · 2024-08-23
> > > >
> > > > We thank the reviewer for many insightful comments. We tried our best to address and incorporate many of the suggestions, as issues such as modalities and legal frameworks are highly important in the context of health data governance. Our additions are likely not comprehensive, as we were also mindful of other reviewers' comments suggesting we focus the paper more on fewer issues (specifically, problems related to dataset duplication on CCPs). Nevertheless, we hope that our additions still convey the importance of these topics to the reader. We would welcome opportunities to discuss these topics further with the reviewer after the discussion period and paper decisions. It seems that more needs to be written about these issues, and perhaps this could open opportunities for collaboration.
> > > >
> > > > > Existing tools and standard for healthcare data The paper does not mention or discuss existing healthcare data standards for ML such as the Fast Health Interoperability Resources (FHIR) that allows standardized data access REST architectures and JSON data formats [1] or specialized Python libraries for radiology/pathology images [2], or DL frameworks for medical imaging [3]; these are just a few examples, you should search for more. If the aim is to improve data quality and practices for MI in ML, then there should be more focus on the existing data standards and software already available but unknown to the broader ML community. Specifically, it would be helpful to point to examples from other papers to detail the challenges with working with clinical data and deploying clinical AI applications.
> > > >
> > > > In "Data governance for healthcare data" of Section 2.2, we rephrased and added "To take into account underrepresented populations and ensure transparency, ML researchers entering medical applications should adhere to established healthcare standards. These include data standards like Fast Health Interoperability Resources (FHIR) [1], which allows standardized data access using REST architectures and JSON data formats. Additionally, they should follow standard reporting guidelines such as TRIPOD (Transparent Reporting of a multivariable prediction model for Individual Prognosis Or Diagnosis) and CONSORT (Consolidated Standards of Reporting Trials, which are now being adapted for ML applications."
> > > >
> > > > We agree that specific software and libraries are necessary for deploying clinical AI applications. In the previous revision, we added MONAI framework to "Initiatives to work on data and improve the data lifecycle" in Section 4. However, the focus of our paper is on available datasets and how they are used. In our view, the problems we encounter we encounter on CCPs are not due to the lack of MI-specific frameworks, but rather because one-size-fits-most frameworks exist, and MI datasets are forced to fit those frameworks regardless. A detailed investigation of software and libraries would, of course, be of interest for future work.
> > > >
> > > > References
> > > >
> > > > [1] Ayaz, Muhammad, et al. "The Fast Health Interoperability Resources (FHIR) standard: systematic literature review of implementations, applications, challenges and opportunities." JMIR medical informatics 9.7 (2021).

---

> > ### Author Rebuttal · Authors · 2024-08-23
> >
> > (continuation of rebuttal - 2/3)
> >
> > > Better differentiating in unique challenges for MI data The paper should better clarify the unique challenges of MI data. How to show that data duplication, missing data, or other quality issues are much worse in MI than CV in general? [...] Otherwise, the arguments for the unique challenges of MI should focus on the challenges of interpreting and annotating MI data, the data dimensionality (3D/4D fMRI), or other domain-specific characteristics such as difficulty in generalization across institutions, etc. More subdomain-specific considerations could be considered such as the computational challenges in processing Whole-Slide Images (WSI) for pathology [6] or the complex preprocessing in radiology and neuroscience such as windowing, registration, and resampling --- along with the reproducibility challenges associated with these preprocessing steps. How to succinctly and effectively convey these challenges to ML practitioners? Providing example images of MI processing may be helpful here.
> >
> > In "Not just “small computer vision”!" of Section 2.1 we already described several characteristics of medical imaging datasets and highlighted their unique properties in terms of anonymization, interdependence and patient demographics. Of course, the modalities and preprocessing themselves are also unique, and we have elaborated on this in the introduction to this section: "The diversity of image modalities and data preprocessing needed for each specific application is vast. For instance 3D images from modalities like MRI can vary significantly depending on the sequence used. For example, brain MRI sequences (T1-weighted, T2, FLAIR, etc.), are designed to emphasize different brain structures, offering specific physiological and anatomical details. Whole-slide images of histopathology are extremely large (gigapixel) images, making preprocessing both challenging and essential for accurate analysis. A crucial part of this process is stain normalization, which standardizes color variations caused by different staining processes, ensuring consistency across slides for more reliable analysis and comparison [6]. We refer interest readers in knowing more about preparing MI data of different modalities for ML for example to [13,14].
> >
> > Nevertheless, the complexity of medical image data above is often reduced to a collection of ML-library-ready images and labels. Yet treating MI datasets as equivalent to benchmark CV datasets is problematic and leads to harmful effects, also termed *data cascades*."
> >
> > > The findings of the study in Section 3 are not detailed enough to show this convincingly. Instead of only mentioning license and availability through CCP/RP, it should quantify the quality and documentation of the data.
> >
> > It would be very important to have a way to (objectively) quantify the level of documentation on CCPs. Although the Datasheets framework [15] does mention documentation, it does not provide guidelines for assessing the quality as high/medium/low quality. In contrast, for data availability and licensing, for example, there is a guide that translates data availability into a score: https://reusabledata.org/criteria. We did, of course, go through the available documentation for the datasets in Table 1, which informed our findings in Section 3, but we chose to include only properties we could annotate unambiguously.
> >
> > A related issue is that while the original dataset might have had documentation, its version on a CCP might not. See for example the cases we already earlier described in Section 3, such as a copy of INBreast with only "I'm just uploading here this data as a backup" as documentation, or copies of MIMIC-CXR with no documentation or reference at all.
> >
> > This is important, so we added a discussion on quantifying documentation quality in "Datasets documentation should be complete and up-to-date" in Section 4. We added: "We note that while we talk about completeness of documentation and the frameworks provide guidelines on what kind of information should be included, it remains unclear how one would quantify that the documentation is, for example, 86% complete in a way that truly reflects the needs of data stakeholders and is not merely a box-ticking exercise."
> >
> > References
> >
> > [6] Cui, Miao, and David Y. Zhang. "Artificial intelligence and computational pathology." Laboratory Investigation 101.4 (2021): 412-422.
> >
> > [13] Willemink, M. J., ... & Lungren, M. P. (2020). Preparing medical imaging data for machine learning. Radiology, 295(1), 4-15.
> >
> > [14] Langlotz, C. P., ... & Kandarpa, K. (2019). A roadmap for foundational research on artificial intelligence in medical imaging: from the 2018 NIH/RSNA/ACR/The Academy Workshop. Radiology, 291(3), 781-791.
> >
> > [15] Gebru, T., ... & Crawford, K. (2021). Datasheets for datasets. Communications of the ACM, 64(12), 86-92.

---

> > > ### Author Rebuttal · Authors · 2024-08-23
> > >
> > > (continuation of rebuttal - 3/3)
> > >
> > >
> > >
> > >
> > > > Social, ethical, and legal challenges of MI data The paper should provide more discussion on the associated ethical and social challenges with working with medical data, starting from data acquisition and governance. Topics of interest include linkage attacks [7], privacy regulations such as HIPPA [8], standards for reporting secondary findings [9], the commodification of health data [10], alternative data governance structures [11], and public participation in data use [12]. More inclusion of these topics would further help distinguish MI challenges from CV data.
> > >
> > > We thank the reviewer for suggesting additional topics for discussion. We agree that these are important and have integrated some of these ideas in  "More insights needed from all people involved" in Section 4. We added the following: "Public participation in data use [12], alternative data sharing, documenting, and governance models [11] are crucial to addressing power imbalances and enhancing data's generation of value as a common good."
> > >
> > > References
> > >
> > > [11] Duncan, Jamie. "Data protection beyond data rights: Governing data production through collective intermediaries." Internet Policy Review 12.3 (2023).
> > >
> > > [12] Ghafur, Saira, et al. "Public perceptions on data sharing: key insights from the UK and the USA." The Lancet Digital Health 2.9 (2020): e444-e446.

---

### Official Review · Reviewer_J8T9 · 2024-07-24
**Narrow analysis of openly available medical imaging metadata**

**Rating:** 6
**Confidence:** 4
**Clarity:** Yes. I understood the paper easily an…

**Review:**

I thought the author's summary of the characteristics of medical imaging datasets and why they are "not just "small computer vision"" was very strong. I felt they made a clear argument for why MI is different to other forms of ML.

I appreciated learning about the duplicate datasets on CCPs. In retrospect I'm not surprised but I felt that was an important point to draw the community's attention to. The rest of the findings are also unsurprising - missing licenses, lack of persistent identifiers and limited structured summaries / coherent metadata are well known challenges.

I also agree with the team's recommendations: I felt their points about the commons based governance model and the need to have more insights from all people involved in the data lifecycle were very strong.

Where I felt this paper did not meet its potential is the lack of analysis of proprietary data. They state that "Companies providing proprietary MI datasets or labeling services handle challenges such as licensing, documentation, and data quality, offering greater customization and flexibility. Such proprietary datasets remain unaffected by the mentioned challenges [82, 113]." I find this last statement very hard to believe. We just can't judge how well the proprietary datasets meet data quality standards. The authors say "Historically, MI datasets were often proprietary, limited to specific institutions, and held in private repositories." My perception is that this remains the vast majority of MI datasets that are used for ML applications. As a result this paper is looking at a very small set of MI and although I think their recommendations would apply to proprietary datasets too, there's no evidence for that point in the analyses they've conducted.

Ultimately there's an interesting finding that's mentioned in the title - about the duplicated datasets available online - but the rest of the paper has a broader and more generic message.

**Strengths:**

I thought the paper was well written with the background and recommendations particularly clear. Understanding the biases in MI and how MI machine learning applications are different to other types of ML is likely to be of interest to a large number of ML / AI researchers.

**Additional Feedback:**

No additional feedback. I'm sorry I sound a little like the reviewer that wanted you to write a different paper. I think with a few edits you can make your point a little more clearly. I'm looking forward to understanding how the work brings additionality to the field :)

**Correctness:**

The claims made in the submission are clear and evidenced.

I note here that the analyses are limited to openly available MI which makes generalising the points to all MI analyses quite hard.

**Documentation:**

The extracted documentation parameters from Kaggle and HuggingFace are listed in in the supplementary material.

There is no code or open data available to reproduce the findings.

**Ethics:**

No ethical concerns.

**Limitations:**

The authors are clear about the methods they followed and the limitations, with the exception of the lack of critical consideration of their analysis of only openly available MI and how that data relates to proprietary MI datasets.

**Opportunities For Improvement:**

I really like the title - I was excited to read the paper based on it! But I felt as I was reading that the title only related to one finding in amongst others that I've seen before.

I wonder if you can re-jig the paper to either focus more on those duplicated datasets and curate your recommendations to intervening with them to prevent / remove duplicates. Or you could maybe adjust the title (I'm not totally sure if that's allowed so please accept my apology if not!) to make clear that the discussion is more focused on the challenges around using openly available MI data?

Your conclusion was that your "observations reveal that the existing CCP governance model falls short of maintaining the necessary quality standards and recommended practices for sharing, documenting, and evaluating open MI datasets. Our recommendations aim to promote better data governance in the context of MI datasets to mitigate these risks and uphold the reliability and fairness of AI models in healthcare." Why then do you introduce the analysis of CV and NLP data? The comparison between MI and CV / NLP doesn't seem to be a leading finding in the conclusion? Nor relating to the title? Is the purpose of the paper to understand the challenges of CCP governance or the comparison of MI to other domains?

As I said above, the biggest challenge for me is that this is not really a study of MI analyses but of a very small subset of MI analyses that use open data. There's no comparison with the proprietary data - except one sentence assuming that all closed data is high quality. (I think it probably is in many ways, but it could also be small and homogenous as you state!) I'm not really sure how you improve this aspect but I think maybe there are some caveats you could fit into the framing of the work to help guide the reader through to the conclusions in an easier to follow path.

**Relation To Prior Work:**

I thought the background summary was very clear and situated the paper well in questions relating to open MI data quality. I didn't quite understand how novel this work was compared to other investigations in the literature.

**Summary And Contributions:**

[Please excuse my quoting from the paper for this summary - I tried rephrasing and it just sounded clunky! I hope you don't mind me using the authors words to describe their methods and findings.]

This paper investigates publicly available data sets relating to medical imaging (MI). They analyse "dataset distribution in community contributed platforms (CCPs) such as Kaggle and HuggingFace (HF), and regulated platforms (RP) such as Tensorflow (TF), Keras, and PyTorch". They find "vague licenses, lack of persistent identifiers and storage, duplicates and missing metadata".

The team "investigate dataset sharing, documentation, and hosting practices for the 30 most cited CV, NLP, and MI datasets by selecting top-10 datasets for each field by querying Papers with Code with “Images”, “Text”, and “Medical” in the Modality field." They find that MI datasets are rarely available on RPs but are sometimes available on CCPs, whereas there are many NLP and CV datasets available through regulated platforms. The lack of license is comparable across the different focus areas (70% missing license for CV, 50% for MI).

The title of the paper refers to the finding that there are multiple duplicates of open MI datasets. For example there are 27 datasets derived from the international skin imaging collaboration (ISIC) data on HuggingFace and 640 on Kaggle. They can be different due to the application of preprocessing steps (cropped images) or the provision of additional annotations.

Another finding is that HuggingFace provides more structured and complete documentation compared to Kaggle. However the authors point out that it is possible to game the system to get a high usability score on Kaggle - which is related to the "well-documented" tag. For example data providers can set the update frequency to "never" and providing vague information about provenance such as "uses internet sources".

The authors provide a discussion about the "asymmetry between open data and proprietary datasets" - although they do not evaluate any proprietary datasets - and recommend that "access to open datasets should be predictable, compliant with open licensing, and persistent". They further recommend that "open datasets should be evaluated against the context of real-world use" and "datasets documentation should be complete and up-to-date". They advocate for "initiatives to work on data and improve the data lifecycle" and recommend that "CCPs could benefit from commons-based governance" and that there need to be "more insights from all people involved".

---

> ### Author Rebuttal · Authors · 2024-08-16
>
> We would like to thank Reviewer J8T9 for their insightful evaluation and comments.
>
> We are encouraged by the reviewer’s recognition of our strong summary of the characteristics that makes MI different from other ML applications, and acknowledge the importance of highlighting issues like lack of persistent identifiers, vague licenses and dataset duplication on CCP, Additionally, the reviewer agreed with our recommendations and highlighted the strength of the points about the “commons-based governance model” , the need to have more insights “from all people involved in the data lifecycle” and the clarity of the background.
>
> We understand the concerns of the reviewer. We clarify them below.
>
> > Your conclusion was that your "observations reveal that the existing CCP governance model falls short of [...] Our recommendations aim to promote better data governance in the context of MI datasets to mitigate these risks and uphold the reliability and fairness of AI models in healthcare." Why then do you introduce the analysis of CV and NLP data? [...] Is the purpose of the paper to understand the challenges of CCP governance or the comparison of MI to other domains?
>
> Thank you for the question. This is a good point, and we have the chance to clarify what may have been misunderstood. Our conclusion is for medical imaging datasets because our focus is on dataset management recommendations for MI datasets, and their unique challenges on the CCP data governance framework.
>
> As reviewer XUa5 highlighted “There is no related work. This work is first of its kind”. We presented related work on dataset management practices on general machine learning datasets (CV /NLP) and highlighted the unique characteristics of medical imaging datasets with respect to other ML applications. The comparison of MI to CV and NLP is justified because MI is often inspired by CV and NLP, and where the topics of CCPs and data governance have received more attention recently. Moreover, MI is used as an additional application in many CV papers.
>
> In our work, we show how specific deficiencies in the studied MI datasets (i.e. vague licenses,  lack of persistent identifiers and storage, duplicates, and missing metadata) affect their overall quality. We then investigate how these deficiencies are reflected in the practices of CCPs, and how CCPs could benefit from adopting governance actions inspired by the legacy of commons-based governance.
>
> We have rearranged the presentation of Section 3. Findings to make the paper more comprehensive. In particular, we updated the "Study setup" to clarify our goal and comparison with CV and NLP,  and highlighted the shared (with CV and NLP) and unique issues for MI datasets on CCPs.
> * In "Study setup", we added "We aim to promote better practices in the context of MI datasets. We include in the comparison CV and NLP because MI is often inspired by these other ML areas, and is where data governance have received more attention recently."
> * We have rearranged the order of the paragraphs to present first "Lack of persistent identifiers and storage" and we have merged this paragraph with "Missing licenses on CCPs".
> * In "Duplicate datasets and missing metadata on CCPs", we rephrased and added "The lack of documentation for all ML datasets hampers tracking their usage, potentially violates sharing agreements or licenses, and hinders reproducibility. Moreover, due to the characteristics of MI datasets, models trained on datasets missing metadata could result into overoptimistic performance due to data splits mixing patient data, or bias [2] or shortcuts [3,4]."
>
> > There's no comparison with the proprietary data - except one sentence assuming that all closed data is high quality [...] maybe there are some caveats you could fit into the framing of the work to help guide the reader through to the conclusions in an easier to follow path.
>
> We are indeed unfortunately not able to analyze proprietary data due to lack of access. We do have some experiences with this topic, but as these accounts are anecdotal we only discuss this in Section 4. Discussion and not in Section 3. Findings. We do however cite work that has investigated proprietary datasets and back up our statements with corresponding literature.
>
> >  I really like the title - I was excited to read the paper based on it! But I felt as I was reading that the title only related to one finding in amongst others that I've seen before. [...]
>
> We understand the point and appreciate the suggestion. We have restructured the presentation of the findings (as pointed out above) and added one more example of dataset duplication (BRATS dataset). We will try to add more datasets before the discussion period is over. We added "On Kaggle, we found 350 datasets related to BraTS (Brain Tumor Segmentation) [1]. BraTS datasets originated from challenges (2012-2022). These datasets are hosted at different websites and we couldn't retrieve their total size to run a similar comparison to ISIC."
>
> References
>
> [1] Menze, B. H., ... & Van Leemput, K. (2014). The multimodal brain tumor image segmentation benchmark (BRATS). IEEE transactions on medical imaging, 34(10), 1993-2024.
>
> [2] Larrazabal, ... & Ferrante, E. (2020). Gender imbalance in medical imaging datasets produces biased classifiers for computer-aided diagnosis. Proceedings of the National Academy of Sciences, 117(23), 12592-12594.
>
> [3] Oakden-Rayner, ... & Ré, C. (2020, April). Hidden stratification causes clinically meaningful failures in machine learning for medical imaging. In Proceedings of the ACM conference on health, inference, and learning (pp. 151-159).
>
> [4] Winkler, J. K., ... & Haenssle, H. A. (2019). Association between surgical skin markings in dermoscopic images and diagnostic performance of a deep learning convolutional neural network for melanoma recognition. JAMA dermatology, 155(10), 1135-1141.

---

> > ### Comment · Reviewer_J8T9 · 2024-08-31
> > **A clear and constructive rebuttal, key limitation of focusing only on open MI data remains**
> >
> > With my apologies for such a late reply (at the end of the summer vacation!) I acknowledge the response from the authors.
> >
> > Although I think the clarifications are very good and I really appreciate how constructive the authors have been, I think the major limitation of not being able to compare to proprietary medical imaging datasets and models make the broad interpretations from this paper difficult to consider to be of high applicability to the MI field. I think my scores remains appropriate so I don't intend to change them.
> >
> > I will also note that I think the engagement and discussion with Reviewer `Lyfj` is exceptional. I agree with `Lyfj`'s comments whole heartedly. In particular I'd like to try to incorporate their points in [this comment](https://openreview.net/forum?id=X4KImMSIRq&noteId=it29qwudI6) into mine above. My focus is not narrowly focused on "missing access to proprietary data" it is rather that the whole governance and data processing steps for the vast majority of MI analysis are not represented by teams using data available on open platforms.

---

### Official Review · Reviewer_Pct9 · 2024-07-27
**Important points raised**

**Rating:** 7
**Confidence:** 4
**Correctness:** Yes
**Clarity:** The paper is easy to follow and well …

**Review:**

Pertinent issues raised and analyzed by the authors which researchers (including myself) have faced. The issues raised in the paper, I hope, will resonate with the deep learning in the medical imaging community and help towards sensitizing data contributors and CCPs and take concrete actions.

**Strengths:**

The issues raised in the paper is of wide relevance to the broader Medical Imaging community.

**Additional Feedback:**

- In Figure 2, what does each block in the right diagram represent? Are each blocks a single instance of ISIC-derived dataset on Kaggle with block size representing derived dataset size?

- Typos: Line 136 Ther -> There

- We have had similar experience when exploring publicly available intraoral lesion datasets available in Kaggle with documentations along the lines of “You can try and practice on this dataset but We're not 100% sure about reliability.” At least some of these datasets also had train/test leakage issues. We ended up not using most of these public datasets or had to manually filter them for our use case.
 Additionally, Datasets across platforms like Mendeley Data, figshare require licensing, readme and usage metrics but, as said in the paper, could use some commons-based governance, quality indication, metadata etc.

**Documentation:**

N/A

**Opportunities For Improvement:**

Not necessarily a limitation but additional examples of dataset duplication, versioning and alteration might help make the paper more comprehensive.

**Relation To Prior Work:**

Yes, the relation to prior work is clearly stated.

**Summary And Contributions:**

The paper raises issues in dataset management and documentation in community-contributed platforms (CCPs). Use-case examples are shown on how duplication and alteration of datasets with improper documentation exists and that data cards are not always enforced by CCPs. Various good practices are highlighted.

---

> ### Author Rebuttal · Authors · 2024-08-16
>
> We would like to thank Reviewer Pct9 for their valuable feedback.
>
> We are grateful for the acknowledgment of our paper's significance to the wider medical imaging community and for recognizing the value of the practices we presented.
>
> We have integrated the suggestions proposed by the reviewer below.
>
> > Not necessarily a limitation but additional examples of dataset duplication, versioning and alteration might help make the paper more comprehensive.
>
> We already added an additional dataset (BRATS) to the duplication analysis experiments. We will try to add more datasets to this analysis before the discussion period ends. We added "On Kaggle, we found 350 datasets related to BraTS (Brain Tumor Segmentation) [1]. BraTS datasets originated from challenges (2012-2022). These datasets are hosted at different websites and we couldn't retrieve their total size to run a similar comparison to ISIC."
>
> We have also rearranged the presentation of Section 3. Findings to make the paper more comprehensive. In particular, we updated the justification of comparing against non-MI datasets in the "Study setup", and highlighted the shared (with CV and NLP) and unique issues for MI datasets on CCPs.
>
> * In "Study setup", we added "We aim to promote better practices in the context of MI datasets. We include in the comparison CV and NLP because MI is often inspired by these other ML areas, and is where data governance have received more attention recently."
> * We have rearranged the order of the paragraphs to present first "Lack of persistent identifiers and storage" and we have merged this paragraph with "Missing licenses on CCPs".
> * In "Duplicate datasets and missing metadata on CCPs", we rephrased and added "The lack of documentation for all ML datasets hampers tracking their usage, potentially violates sharing agreements or licenses, and hinders reproducibility. Moreover, due to the characteristics of MI datasets, models trained on datasets missing metadata could result into overoptimistic performance due to data splits mixing patient data, or bias [2] or shortcuts [3,4]."
>
> > In Figure 2, what does each block in the right diagram represent? Are each blocks a single instance of ISIC-derived dataset on Kaggle with block size representing derived dataset size?
>
> Exactly. We have clarified the caption of Figure 2. The current caption reads "Representation of the storage size for ISIC (skin lesion) datasets. While the ISIC website hosts a total of 38 GB of data (left), on Kaggle there are a total of 640 datasets related to ISIC (some preprocessed, other with additional annotations), that sum up to 2.35 TB of data (right). Each block on the (right) represents a single instance of ISIC-derived dataset on Kaggle. Block size represents dataset size. Data was retrieved on May, 15 2024."
>
> References
>
> [1] Menze, B. H., Jakab, A., Bauer, S., Kalpathy-Cramer, J., Farahani, K., Kirby, J., ... & Van Leemput, K. (2014). The multimodal brain tumor image segmentation benchmark (BRATS). IEEE transactions on medical imaging, 34(10), 1993-2024.
>
> [2] Larrazabal, A. J., Nieto, N., Peterson, V., Milone, D. H., & Ferrante, E. (2020). Gender imbalance in medical imaging datasets produces biased classifiers for computer-aided diagnosis. Proceedings of the National Academy of Sciences, 117(23), 12592-12594.
>
> [3] Oakden-Rayner, L., Dunnmon, J., Carneiro, G., & Ré, C. (2020, April). Hidden stratification causes clinically meaningful failures in machine learning for medical imaging. In Proceedings of the ACM conference on health, inference, and learning (pp. 151-159).
>
> [4] Winkler, J. K., Fink, C., Toberer, F., Enk, A., Deinlein, T., Hofmann-Wellenhof, R., ... & Haenssle, H. A. (2019). Association between surgical skin markings in dermoscopic images and diagnostic performance of a deep learning convolutional neural network for melanoma recognition. JAMA dermatology, 155(10), 1135-1141.

---

### Author Rebuttal · Authors · 2024-08-16

We would like to sincerely thank the reviewers for their thorough evaluation of our manuscript and for their insightful comments and suggestions. We greatly appreciate the time and effort you invested in providing this valuable feedback, which has significantly contributed to the improvement of our work.

We found many helpful suggestions and questions in the reviewers' comments and addressed each of them in the individual threads. Here, we summarize the main revisions of the paper conducted in response to the reviews.

The main changes are:
* In Section 3. Findings, we explained more about our motivations to focus on MI but contrast it with CV and NLP. We also restructured the section to have more emphasis on the duplication part of the issues with the datasets, and highlighted the shared issues (with CV and NLP) and the unique ones for MI datasets on CCPs.

* We already added an additional dataset (BraTS) to the duplication analysis experiments. We will try to add more datasets to this analysis before the discussion period ends.

* We extended our discussion on "Initiatives to work on data and improve the data lifecycle" including some more initiatives to improve medical dataset management practices, like project MONAI, as well as multi-dataset benchmarks like MedMNIST.

* We fixed any missing definitions, typos, etc.

Let us know if you have any concerns left after our response, and if we have missed any comment. We would be happy to discuss any further questions and comments you may have.

We thank everyone involved in reviewing the paper for their time.

---

### Decision · Program_Chairs · 2024-09-26

**Decision:**

Accept (Poster)

**Comment:**

This work addresses important and often overlooked issues around dataset documentation and governance, with an emphasis on medical imaging datasets. Raising and discussing these issues, especially providing pointers to better practices, is an important contribution to this research community. This is the key strength of this work, as acknowledged by all of the reviewers. The exposition on how medical imaging datasets differ from other types of data typically used in ML research is also valuable for researchers less familiar with medical imaging. The case study on the proliferation of ISIC helps to demonstrate some of the points raised.

Where the work is more lacking is 1) clarity of messaging and scope, and 2) quantitative substantiation of findings.

As highlighted by several of the reviewers, only ten medical imaging datasets are considered in Table 1, with the other 20 datasets coming from non-MI domains, and the issue in the title ("Copycats...") is only explored in a case study for ISIC, and BRATS as added during the discussion period. The inclusion of non-MI datasets in Table 1 is not especially compelling - the authors argue "We include in the comparison CV and NLP because MI is often inspired by these other ML areas, and is where data governance have received more attention recently", but as highlighted by Reviewer Lyfj, Table 1 suggests non-MI datasets may actually have worse data governance than MI, with license information more readily available for the MI datasets. It then raises the question of whether the purpose of this work is to compare MI with non-MI from a data governance perspective, which makes the scope of the work less clear, an issue perhaps compounded by the title referring to one phenomenon (dataset duplication) explored in a case study on MI datasets only. To quote Reviewer [J8T9], "Ultimately there's an interesting finding that's mentioned in the title - about the duplicated datasets available online - but the rest of the paper has a broader and more generic message".

It seems there is a missed opportunity to provide a more nuanced view of the state of MI datasets given their unique properties (as outlined in Section 2.1), given the authors seemingly did some additional analysis on the nature of metadata available for the given datasets (e.g. as alluded to in lines 233-255, and other sections). For example, is it possible to know how often splits are defined on a subject level for the MI datasets studied?

If there are additional data tables collected by the authors during this study, providing them would help to justify some of the claims in the paper, such as "we find composition and collection process are the two fields most represented; motivation of the creation of the dataset is rarely included in the general description of the dataset; information about preprocessing/cleaning/labeling or about uses is usually missing" - what do "rarely", "usually" mean here?

As it, is the paper falls somewhere between a position paper and a systematic review, where as a systematic review I would find it lacking. However, given the importance of data governance for this community, and the growing relevance of medical imaging within machine learning, the contribution of this work in raising concerns and advocating for better practices is sufficient to argue for its inclusion in the NeurIPS Datasets and Benchmarks track. I strongly recommend the authors to include the additional analyses (e.g. on BRATS) and discussion points raised during the discussion period to further strengthen the paper.